# An auto-inhibited state of protein kinase G and implications for selective activation

Rajesh Sharma[1], Jeong Joo Kim[1†], Liying Qin[1], Philipp Henning[2], Madoka Akimoto[3], Bryan VanSchouwen[3], Gundeep Kaur[1], Banumathi Sankaran[4], Kevin R MacKenzie[1,5]*, Giuseppe Melacini[3], Darren E Casteel[6], Friedrich W Herberg[2], Choel Kim[1,7]*

[1]Department of Pharmacology and Chemical Biology and Center for Drug Discovery, Baylor College of Medicine, Houston, United States; [2]Department of Biochemistry, University of Kassel, Kassel, Germany; [3]Department of Chemistry and Chemical Biology, McMaster University, Hamilton, Canada; [4]Molecular Biophysics and Integrated Bioimaging, Berkeley, United States; [5]Department of Pathology and Immunology and Center for Drug Discovery, Baylor College of Medicine, Houston, United States; [6]Department of Medicine, University of California, San Diego, San Diego, United States; [7]Verna and Marrs McLean Department of Biochemistry and Molecular Biology, Baylor College of Medicine, Houston, United States

*For correspondence:
kevin.mackenzie@bcm.edu (KRMacK);
ckim@bcm.edu (CK)

Present address: †Department of Therapeutics Discovery, Amgen Research, Amgen Inc, South San Francisco, United States

**Competing interest:** The authors declare that no competing interests exist.

**Abstract** Cyclic GMP-dependent protein kinases (PKGs) are key mediators of the nitric oxide/cyclic guanosine monophosphate (cGMP) signaling pathway that regulates biological functions as diverse as smooth muscle contraction, cardiac function, and axon guidance. Understanding how cGMP differentially triggers mammalian PKG isoforms could lead to new therapeutics that inhibit or activate PKGs, complementing drugs that target nitric oxide synthases and cyclic nucleotide phosphodiesterases in this signaling axis. Alternate splicing of PRKG1 transcripts confers distinct leucine zippers, linkers, and auto-inhibitory (AI) pseudo-substrate sequences to PKG Iα and Iβ that result in isoform-specific activation properties, but the mechanism of enzyme auto-inhibition and its alleviation by cGMP is not well understood. Here, we present a crystal structure of PKG Iβ in which the AI sequence and the cyclic nucleotide-binding (CNB) domains are bound to the catalytic domain, providing a snapshot of the auto-inhibited state. Specific contacts between the PKG Iβ AI sequence and the enzyme active site help explain isoform-specific activation constants and the effects of phosphorylation in the linker. We also present a crystal structure of a PKG I CNB domain with an activating mutation linked to Thoracic Aortic Aneurysms and Dissections. Similarity of this structure to wildtype cGMP-bound domains and differences with the auto-inhibited enzyme provide a mechanistic basis for constitutive activation. We show that PKG Iβ auto-inhibition is mediated by contacts within each monomer of the native full-length dimeric protein, and using the available structural and biochemical data we develop a model for the regulation and cooperative activation of PKGs.

## Editor's evaluation

This crystal structure of nearly full-length human cGMP-dependent protein kinase Iβ (PKG Iβ) provides convincing new insights into how in the absence of cGMP the activity of the catalytic domain is held in check by intramolecular interactions between both the upstream regulatory cGMP-binding domains and autoinhibitory segment and the catalytic domain, and how cGMP binding to the two cGMP-binding domains can relieve these inhibitory constraints leading to an increase in catalytic activity. The regulatory interactions in PKG Iβ reveal similarities and differences to the way in which the cAMP-dependent protein kinase (PKA) regulatory domain inhibits the PKA catalytic

subunit. The new structure of the activating PKG Iα R177Q CNB-A domain mutant, which resembles a cGMP-bound wild-type CNB-A domain, provides a nice explanation for how this point mutation activates PKG Iα and leads to the development of the TAAD (Thoracic Aortic Aneurysms and Dissections) syndrome.

## Introduction

The second messenger cyclic guanosine monophosphate (cGMP) regulates a myriad of physiological processes including cellular growth, smooth muscle contractility, cardiovascular homeostasis, inflammation, sensory transduction, bone growth, and neuronal plasticity and learning (*Battye et al., 2011*; *Francis et al., 2010*; *Hammond and Balligand, 2012*; *Klinger and Kadowitz, 2017*). In eukaryotes, cGMP-dependent protein kinases (PKGs) transduce intracellular cGMP levels, which change in response to extracellular signals, into phosphorylation of target proteins that control cell activities. PKGs show minimal kinase activity in the apo state but are strongly, cooperatively, and selectively activated by submicromolar concentrations of cGMP. The regulation of PKGs is of fundamental importance to understanding the basis of biological responses to cyclic nucleotides and could reveal therapeutic strategies that would complement current approaches to colon cancer, hypertensive heart disease, pulmonary hypertension, osteoporosis, and chronic pain (*Browning et al., 2010*; *Feil et al., 2003*; *Kalyanaraman et al., 2018*; *Klinger and Kadowitz, 2017*; *Luo et al., 2014*).

PKGs belong to the AGC family of protein kinases and have N-terminal regulatory (R) and C-terminal catalytic (C) domains. The PKG Iα and Iβ isoforms are splice variants with different N-terminal regions of 89 or 104 residues but the same C-terminal 582 residues that form cyclic nucleotide-binding (CNB) and C-domains (*Francis and Corbin, 1994*; *Hofmann et al., 2009*). PKG I induces smooth muscle relaxation by lowering intracellular calcium or activating myosin phosphatase (*Schlossmann et al., 2000*; *Surks et al., 1999*). PKG II is produced from a different gene with 63% identity and regulates ERK activation required for bone growth, and trafficking of cystic fibrosis transmembrane conductance regulator (*Pfeifer et al., 1996*; *Rangaswami et al., 2009*; *Serulle et al., 2007*; *Vaandrager et al., 1998*). Differences between the isoforms are important to their function (*Schlossmann and Hofmann, 2005*).

Much of our understanding of the mechanism of action of PKGs derives from studies of isolated domains. The R-domain comprises a leucine zipper (LZ) domain, a linker region with an auto-inhibitory (AI) sequence, and two cyclic nucleotide-binding (CNB-A and -B) domains (*Figure 1A*). All three PKG isoforms have unique LZ and AI domains. The LZ domains mediate homodimerization and recruit isoform-specific interacting proteins (*Casteel et al., 2010*; *Qin et al., 2015*; *Reger et al., 2014*). The isoform-specific AI sequences contain pseudo-substrate motifs with the target S/T replaced by G or A (RAQGIS in PKG Iα, KRQAIS in PKG Iβ, and AKAGVS in PKG II) that bind the catalytic cleft, blocking substrate access (*Francis et al., 1996*). The CNB-A and -B domains bind cGMP with differing affinities and selectivities (*Huang et al., 2014b*; *Kim et al., 2011*) and are connected by interdomain helices. Each CNB includes a β-subdomain flanked by helical subdomains (*Berman et al., 2005*; *Rehmann et al., 2007*). Cyclic nucleotide pockets comprise the phosphate-binding cassette (PBC), base-binding region (BBR), and capping residue (*Das et al., 2009*). A PBC arginine that is conserved in all CNBs binds a nonbridging oxygen in the cyclic phosphate. A mutation that causes familial Thoracic Aortic Aneurysms and Dissections (TAAD) maps to this arginine of CNB-A (PKG Iα residue 177), and the mutation to glutamine constitutively activates the enzyme (*Guo et al., 2013*). In PKG I, CNB-B includes a BBR arginine that specifically recognizes cGMP over cAMP; CNB-A lacks this arginine and binds either cyclic nucleotide with similarly high affinity (*Huang et al., 2014b*; *Kim et al., 2011*). Capping residue Y351 in CNB-B shields the guanine moiety from solvent and is also referred to as a lid (*Huang et al., 2014b*).

By analogy to PKA, the PKG R-domain is thought to bind via its AI sequence to the C-domain active site to form an auto-inhibited R:C state; binding of cGMP to the R-domain destabilizes this state and allows active kinase to phosphorylate downstream substrates. In contrast to PKG, activation of the PKA R:C state by cAMP dissociates the AI tetramer into separate R and C subunits. In previous work, we showed that a PKG I R-domain comprising two CNB domains but lacking the dimerization domain makes R:R intermonomer contacts through the cGMP bound at CNB-A and at a kink in the interdomain linker between the two CNBs (*Kim et al., 2016*). Disrupting this interface with point mutations

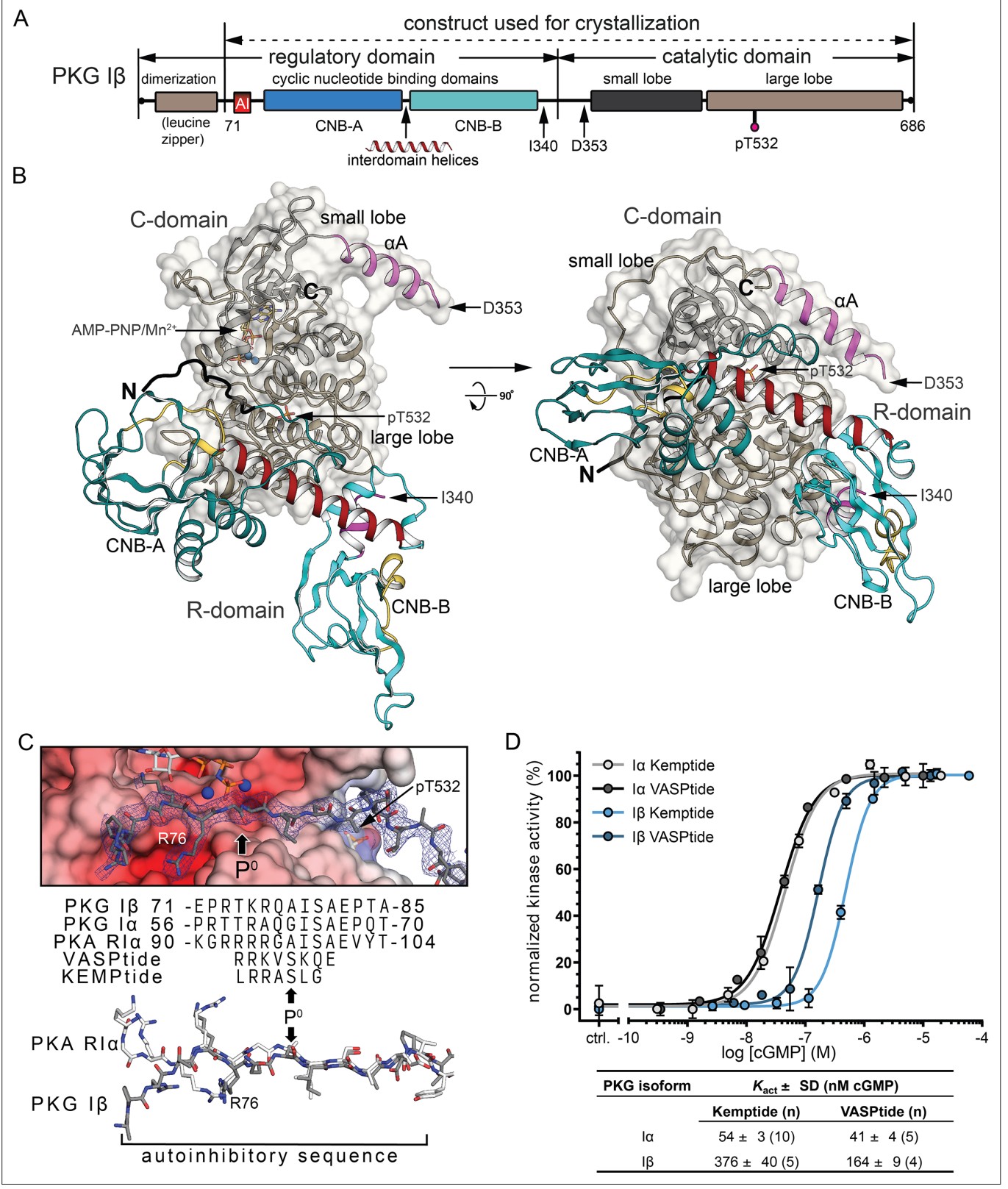

**Figure 1.** Overall structure of the R:C holoenzyme complex. (**A**) Domain organization of PKG Iβ and the construct used for crystallization. AI, auto-inhibitory sequence; CNB, cyclic nucleotide-binding domain. (**B**) Overall structure of the PKG Iβ R:C complex (71–686). The R- and C-domains are shown in cartoons with a transparent surface on the C-domain. N and C termini are labeled. AI is colored in black. CNB-A and -B are colored in teal, phosphate-binding cassette (PBC) in yellow, αA-helix in magenta, and the interdomain helices in red. The αA helix of the C-domain is colored

*Figure 1 continued*

in magenta, the small lobe is in black and the large lobe is in dark tan. The phosphorylated T532 is shown in sticks. The last ordered residue in the R-domain and the first ordered residue in the C-domain are indicated with arrows. The entire C-domain shows clear density in both chains; besides the missing residues discussed in the text, the first two residues at the N-terminus (residues 71–72) of the R-domains and five residues that follow the AI sequence (residues 85–89) in one chain are missing. All structure images were generated using PyMOL (DeLano Scientific). (**C**) AI docking to the active site. Top: AI is shown with electron density ($2F_o - F_c$ at $\sigma = 1.0$). The phosphorylation site ($P^0$) in models and sequences is marked with arrows. Electrostatic potential surface is shown for the C-domain active site. Bottom: Alignment of AI sequences of PKG Iβ, Iα, and PKA RIα with substrates VASPtide and Kemptide. (**D**) Isozyme differences in cyclic guanosine monophosphate (cGMP)-dependent activation of PKG Iα (gray) and Iβ (blue) are revealed by activity measurements, and activation constants ($K_{act}$) vary with substrate. Both isoforms require less cGMP for half-maximal activation using VASPtide (dark gray and dark blue) compared to Kemptide (light gray and light blue). Data points show the mean of duplicates with error bars indication the standard deviation (SD). $K_{act}$ values are given as mean of $n$ measurements ± standard deviation (SD). Additional data related to these fits are presented in *Figure 1—figure supplement 3*.

The online version of this article includes the following figure supplement(s) for figure 1:

**Figure supplement 1.** The R:C dimer captured in asymmetric unit and the R:C monomer alignment with previous structures.

**Figure supplement 2.** The FoXS webserver analysis shows that the theoretically calculated scattering profile from the cis-conformation (red) matches well with the experimentally observed scattering profile (blue) with a $\chi^2 = 1.44$.

**Figure supplement 3.** Isoform-specific differences between PKG Iα and Iβ kinetic parameters.

increases the activation constant in dimeric full-length PKG Iβ, suggesting a role for the domain:domain contacts that mediate this interface in stabilizing the activated state of the holoenzyme (*Kim et al., 2016*) despite the AI sequences tethered in proximity to the C-domains.

Here, we present the first crystal structure of PKG Iβ in which the regulatory domain is free of cyclic nucleotide and adopts an AI state bound to the catalytic domain. The R:C interface reveals specific contacts between the PKG Iβ AI sequence and the active site that help to explain isoform-specific activation constants and the activating effects of phosphorylation in the linker. Comparison with the structures of isolated PKG R- and C-domains shows that the R-domain undergoes local conformational changes and domain rearrangements to assemble a surface that contacts the catalytic core of the C-domain and inhibits its activity. We also present a high-resolution structure of the CNB-A domain with the TAAD mutation, which reveals a closed cGMP pocket. The similarity of this mutant domain to wildtype cGMP-bound structures and its differences with the open pocket in the AI enzyme explain the constitutive activation of the holoenzyme in TAAD. We show that PKG Iβ auto-inhibition is mediated by contacts within each monomer of the native full-length dimeric protein, and using the available structural and biochemical data we develop a model for the regulation and activation of PKGs.

## Results

### Overall structure

The asymmetric unit contains two PKG Iβ 71–686 peptide chains (see *Figure 1A* for domain structure) without cGMP (*Supplementary file 1*). Structures of the individual domains are similar to those of previously reported isolated domains (*Huang et al., 2014b*; *Kim et al., 2011*; *Qin et al., 2018*). Stretches of ~15 amino acids lack electron density in both chains: the R-domain traces end at residue 341 or 342, and the C-domain traces begin at residue 353 or 355, so how they are connected within the crystal is undetermined (*Figure 1—figure supplement 1A*). These residues are helical in structures of the isolated CNB-B domain and of the C-domain (*Figure 1—figure supplement 1B*), but show a high rate of backbone amide hydrogen/deuterium exchange in full-length PKG Iβ, whether it is AI or activated (*Chan et al., 2020*). The shortest connecting path in the crystal (about 15 Å) could be spanned by a kinked helix to give a dimer inhibited in trans, with the R-domain of one monomer bound to the C-domain of the other, while the alternative path for a cis-inhibited dimer could be spanned by an extended loop (about 35 Å) (*Figure 1—figure supplement 1A*). In solution, however, this construct is monomeric, and its experimental radius of gyration from small angle X-ray scattering is in good agreement with the calculated radius of gyration for a monomer formed by the alternate topology (*Figure 1—figure supplement 2* and *Supplementary file 2*).

In the PKG Iβ 71–686 AI complex, the two CNB domains contact the large lobe of the C-domain while the AI sequence docks to the C-domain active site (*Figure 1C*). The C-domain binds AMP-PNP:$Mn^{2+}$ at the ATP site with the DFG motif adopting an 'in' conformation. Both CNB-A and CNB-B

PBCs adopt 'open' conformations with respect to the β-subdomains, similar to previous CNB domain structures without cyclic nucleotide (*Boettcher et al., 2011*; *Kim et al., 2007*; *Lu et al., 2019*). The overall structure is very similar to the PKA RIα:Cα heterodimer showing square root of the average of squared errors (RMSD) values of 1.95 Å for 546 shared Cα atoms (*Kim et al., 2007*; *Minami et al., 2013*), and it is similar to the CNB-A/B and the catalytic domains of monomeric *Plasmodium* PKG showing RMSD values of 2.91 Å for 504 shared Cα atoms (*El Bakkouri et al., 2019*). R:C contacts made by CNB-A use residues that are well conserved between PKG Iβ and PKA, while those made by CNB-B (with the activation loop) and by the AI (with the active site) are less well conserved. The differences in cGMP activation between PKG Iα and Iβ (*Figure 1D*) are partly explained by AI:active site contacts.

## The PKG Iβ R:C interface

In the R:C complex we present here, the interdomain linker between the tandem CNB domains forms a single helix that allows the two CNB domains to clamp around the C-domain. We describe the PKG Iβ R:C contacts in terms of four docking subsites (*Figure 2* and *Supplementary file 3*), following the precedent of the PKA RIα:Cα heterodimer (*Kim et al., 2007*). At the first site, the R-domain AI sequence occupies the C-domain active site cleft (*Figures 1C and 2B*). At the second site, the helical subdomain of CNB-A, including PBC-A and N3A$^A$, combines with the AI to form a surface that binds the αG helix of the C-domain (names for C-domain elements follow the established convention for kinases *Johnson et al., 2001*, and names for CNB elements follow the convention from *Berman et al., 2005*; *Kornev et al., 2008*). At the third site, the interdomain helices and the αA helix of CNB-B shield the C-domain activation loop from solvent, and at the fourth, the αB helix of CNB-B docks to an S-shaped loop (residues 610–625) between the αH and αI helices.

At site 1, the AI sequence docks to the active site cleft of the C-domain through polar and nonpolar interactions (*Figures 1C and 2B*). Residues I79, S80 and A81, which are common to PKG Iα and Iβ, form a three-residue antiparallel β-sheet with the $P + 1$ loop of the C-domain. PKG Iβ-specific residues R73 ($P$ - 5), K75 ($P$ - 3), and R76 ($P$ - 2), which are critical to auto-inhibition (*Francis et al., 1996*), interact with the αD and αF (*Figure 2B*). The pseudo-substrate $P + 1$ residue, I79, docks to a hydrophobic pocket formed by F533, P537, and Y582 and contacts the aromatic side chain of Y188 of PBC-A (*Figure 2B*). E82 makes a salt bridge with R209 of the R-domain αB helix and positions it to interact with N189 at the helical tip of PBC-A helping to organize site 2 of the R-domain.

Compared to the PKA RIα:C complex (*Kim et al., 2007*), PKG Iβ has fewer positively charged residues in the AI sequence and shows fewer hydrogen bonds at site 1 (*Figure 1C*); modeling the PKG Iα sequence suggests that it would make even fewer stabilizing contacts. Mapping the PKG Iα $P$ - 5, $P$ - 3, and $P$ - 2 residues T57, R59, and A60 (*Figure 1C*) onto the PKG Iβ AI structure suggests the loss of two salt bridges with the active site at $P$ - 2 and of nonpolar interactions with the αD helix at $P$ - 5. These differences could weaken the R:C interface and may contribute to the lower activation constant and higher basal activity in PKG Iα (*Figure 1D* and *Figure 1—figure supplement 3*).

At site 2, the C-domain αG helix provides a nonpolar docking surface for PBC-A and N3A$^A$ of CNB-A similarly as in the PKA RIα:C complex (site 2 in *Figure 2B*). A cluster of nonpolar residues including M579 and Y582 in the αG helix docks to the helical tip of PBC-A (L184, L187, and Y188) that adopts an 'open' conformation with respect to the β-subdomain (site 2 in *Figure 2B*). The rest of the αG-helix docks with the $3_{10}$-helix of N3A$^A$.

At site 3, the C-domain activation loop contacts the R-domain at a linker C-terminal to the AI sequence, at an interdomain helix, and at the αA helix of CNB-B. These interactions are unique to PKG Iβ compared to the PKA RIα:C complex. Linker residue P83 stacks with the H420 side chain, which donates a hydrogen bond to the phosphate moiety of pT532 and shields it from solvent, and the T84 side chain hydroxyl and backbone amide donate hydrogen bonds to the phosphate of pT532, which is held in place by two hydrogen bonds from the R498 side chain and one from the T530 side chain (right zoom-in panel of site 1 in *Figure 2B*). Replacing the analogous PKG Iα phosphorylatable threonine (T517) with alanine results in loss of kinase activity, while replacing it with glutamate retains basal activity, indicating that phosphorylation at this site of the activation loop is required for catalysis (*Feil et al., 1995*). The contacts that we report here between the R-domain and this important phosphorylated threonine should help stabilize R:C interaction and thus favor auto-inhibition.

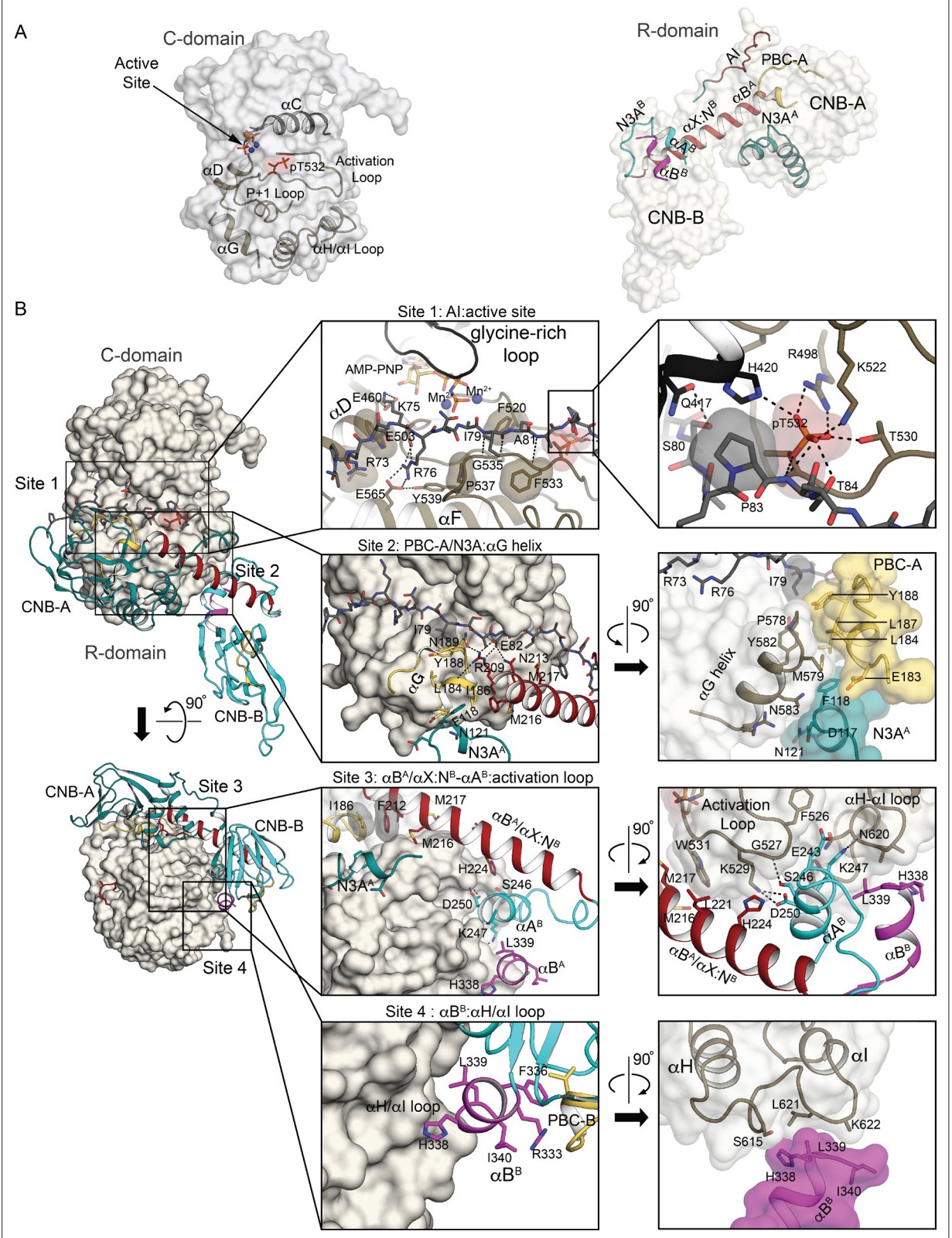

**Figure 2.** Interactions at the R:C interface. (**A**) Key structural elements that form the R:C interface. The C- and R-domains (left and right) are displayed in isolation as transparent surface with key elements labeled and shown as cartoons using the same color scheme as in *Figure 1B*. (**B**) Detailed R:C interactions with the C-domain presented as a surface and the R-domain portrayed in cartoon using the same color scheme. Left two panels: Two orthogonal views of the AI complex with outlines identifying the regions discussed in the text as interaction sites 1 through 4. Middle four panels:

*Figure 2 continued on next page*

*Figure 2 continued*

Zoomed-in views of the four interaction sites with key residues labeled and shown as sticks, and hydrogen bonds shown as dotted lines. Right four panels: Further zoomed-in or rotated views of the four interaction sites. Detailed R:C contacts are summarized in **Supplementary file 3**.

This part of the structure explains why phosphorylation of S80 activates PKG Iβ (**Smith et al., 1996**). The S80 side chain hydroxyl donates a hydrogen bond to the A81 carbonyl oxygen, receives a hydrogen bond from Q417 of the αC helix, and is 3.5 Å from the αC-helix H420 side chain that contacts pT532 (**Figure 2B**). Phosphorylated S80 could not be sterically accommodated and would experience charge repulsion with the phosphorylated activation loop and αC helix of the C-domain. At the analogous position in PKG Iα, the phospho-mimetic mutation S65D constitutively activates **Busch et al., 2002**; this mutation can be accommodated in our structure only by adjusting the Q417 rotamer away from the AI and pointing the S65D rotamer away from the C-domain. We note that mutations S65A in PKG Iα and S80A in PKG Iβ both raise basal kinase activity twofold (**Busch et al., 2002**), presumably due to loss of the S80 hydroxyl contacts with Q417 and A81 and the associated destabilization of the R:C interface.

The interdomain helices and the CNB-B αA helix shield the rest of the activation loop from solvent (left zoom-in panel of site 3 in **Figure 2B**). The side chains of interdomain residue M216, M217, and H224 dock with the side chains of K529 and W531 of the activation loop (right zoom-in panel of Site 3 in **Figure 2B**). The side chain of E243 in the CNB-B αA helix contacts F526 and G527 of the tip of the activation loop while the side chain of D250 forms a salt bridge with the K529 side chain (right zoom-in panel of Site 3 in **Figure 2B**). This region differs substantially from previously reported PKG structures: in the cGMP-bound structure of the R-domain (**Kim et al., 2016**), the interdomain linker consists of bent helices that make R:R intermonomer contacts.

At site 4, the loop between the αH and αI helices of the C-domain interacts with the αB helix of CNB-B in the R-domain similarly as in the PKA RIα:C complex (**Figure 2A**). The end of the αB helix docks to an S-shaped loop between the αH and αI helices (**Figure 2B**). Because the CNB-B αB helix is tightly coupled to the PBC, conformational changes accompanying cGMP binding can disrupt the above interactions and enable activation.

## C-domain architecture and comparison with PKA

The PKG Iβ C-domain in the AI complex is similar to a structure of the isolated C-domain (**Qin et al., 2018**), showing an RMSD of 0.39 Å for 322 Cα atoms that exclude the glycine-rich loop, which is displaced by up to 3.5 Å in the isolated C-domain to accommodate a bound inhibitor (**Figure 3— figure supplement 1**). In the AI PKG Iβ C-domain, the glycine-rich loop contacts bound AMP-PNP, activation loop T532 is phosphorylated, and the DFG motif adopts an 'in' conformation with D517 coordinating $Mn^{2+}$, so the pseudo-substrate is bound much as true substrates are expected to bind for phosphorylation. Differences in the orientations and contacts of the αA helix and the C-terminal tail of PKG compared to PKA may be important to understanding how signaling pathway cross-talk is minimized and how the mechanisms of activation differ (**Figure 3A**). In PKG, the αA helix interacts only with the small lobe, whereas in PKA, the αA helix interacts mainly with the large lobe. The PKG αA helix is linked to the adenosine pocket via the E363:R438 salt bridge and F367:Y440 packing interaction (**Figure 3B**). The significance of this difference is unclear, since residues that precede the αA helix in the PKG AI complex are not ordered and show no density (residues 330–353).

Compared to the PKA C subunit, PKG Iβ has a two residue longer activation loop connecting αA and β1. Superimposing the PKG and PKA structures (excluding the loops) shows that the longer PKG activation loop is accommodated by small PKG R-domain residues E243 and S246, whereas the PKA activation loop contacts larger PKA RIα residues W260 and L263 at the homologous positions (**Figure 3C**). We infer that PKA RIα binding to the PKG C-domain would cause the extended tip of the activation loop of PKG I to clash with the CNB-B A-helix of PKA RIα (**Figure 3C**). These sequence differences, and those of the AIs (**Figure 1C**) discussed above, likely arose from evolutionary pressure to minimize cross-talk between the cyclic nucleotide signaling pathways.

The C-terminal tails of AGC kinases contribute to the overall fold and catalytic function (**Kannan et al., 2007**) and are important for kinase assembly and regulation (**Taylor et al., 2021**). In PKG, the C-terminal tail interacts directly with the kinase core (**Figure 3D**): D665 of the C-terminal tail forms a strong salt bridge with R466 of the αD helix, and D467 stabilizes the position of R466 with a hydrogen

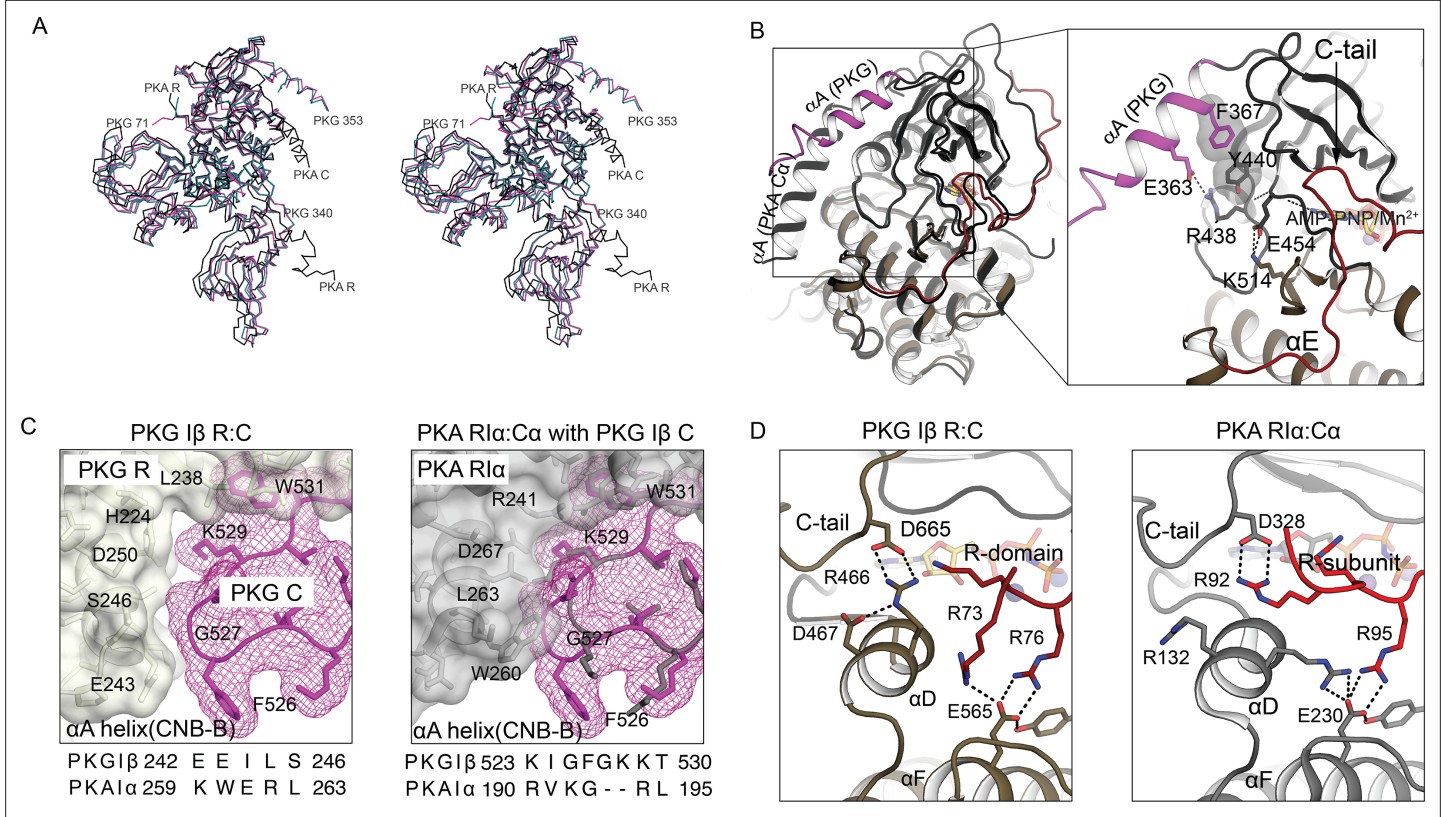

**Figure 3.** Comparison of AI PKG and PKA structures. (**A**) Structural alignment of AI PKG Iβ (red, PDB ID: 7LV3) and PKA Iα (gray, PDB ID: 2QCS) in stereo-view. (**B**) C-domain αA helices are differently positioned in PKG Iβ and PKA. Left: AI PKG Iβ and PKA Iα structures are aligned using Cα atoms of the C-domains Right: Zoom-in view of interactions between the PKG Iβ αA helix and the catalytic core. The interacting residues and bound AMP-PNP:Mn²⁺ (ANP) are shown in sticks and labeled, and hydrogen bonds are shown as dotted lines. The short PKG Iβ αA-β1 loop allows αA to contact the small lobe of the C-domain via a salt bridge (E363:R438) and stacking interactions (F367:Y440) in a way that may be conveyed to the ribose pocket. (**C**) PKG Iβ contacts between the CNB-B αA helix and the C-domain near the activation loop imply that binding of PKA RIα would cause steric clashes. Left: PKG Iβ R:C interface at site 3 with sequences of PKG Iβ and PKA Iα at the CNB-B αA helix aligned. The activation loop is shown in mesh (magenta), the R-domain is shown in transparent surface (gray), and interacting residues are labeled. Right: Aligning AI PKA Iα on AI PKG Iβ using the C-domains (except the activation loop) shows that large PKA RIα side chains L263 and W260 that contact the shorter PKA Cα loop would clash with the larger PKG Iβ loop. Sequence alignment between PKG Iβ and PKA Iα at the activation loop is shown below. (**D**) In auto-inhibition, PKG Iβ and PKA C-terminal tails occupy similar positions but contact different partners. The C- and R-domains (or subunits) are colored in gray and red, respectively. Left: The C-tail of PKG interacts with the C-domain αD helix and the AI linker interacts with the C-domain αF helix. Right: The C-tail of PKA Iα interacts with the AI linker, and both the αD helix and the AI linker interact with the C-subunit αF helix.

The online version of this article includes the following figure supplement(s) for figure 3:

**Figure supplement 1.** Structural alignment of AI PKG Iβ monomer with the isolated PKG I C-domain bound to N46 (PDB ID: 6C0T).

bond. In PKA the C-terminal tail interacts with the R subunit: the PKA homolog of D665 (D328) interacts with R92 (*P* - 5) of the RIα subunit, while R133 of the C subunit αD helix (which is homologous to R466 of PKG Iβ) points toward the active site and interacts with E230 of the αF helix.

## Conformational differences between AI and activated states

Comparing the AI complex with previous structures of PKG and PKA R-domain fragments bound to cyclic nucleotides (*Kim et al., 2016*; *Osborne et al., 2011*) reveals local changes that coordinate opening and closing the PBC in response to cGMP binding (*Berman et al., 2005*; *Kim and Sharma, 2021*; *Taylor et al., 2008*), as well as global changes at the interdomain helices that orient the two CNBs for docking to the C-domain (*Figure 4A,B*). Tracking these changes (and adjacent helices) enables us to develop a rationale for how structural changes induced in the CNBs are coupled to the stabilization of the PKG AI state.

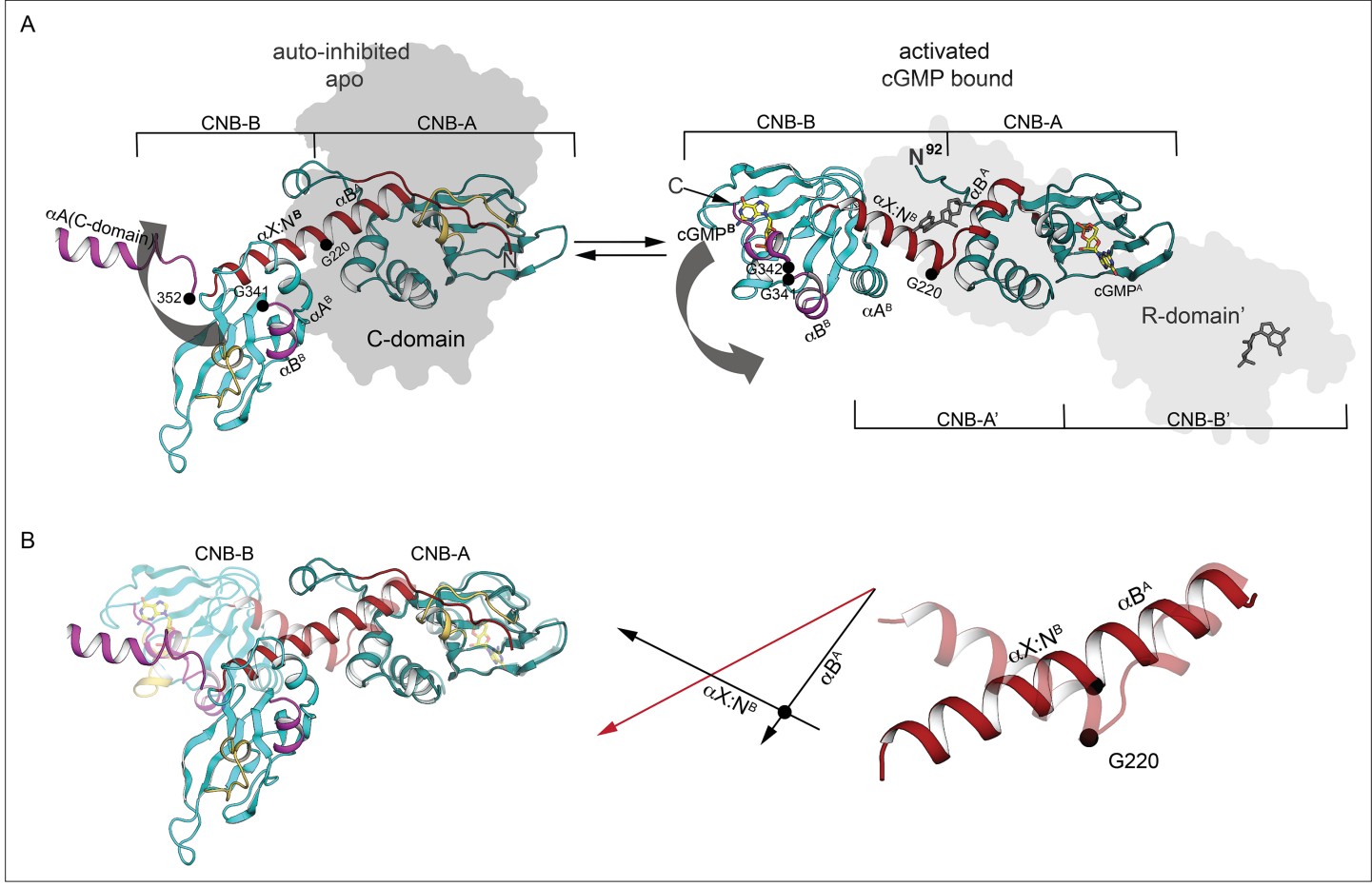

**Figure 4.** Comparison of PKG Iβ R-domain structures in activated and AI states. (**A**) Cartoon of a single R-domain in either (left) the AI R:C state, bound to the C-domain depicted as a gray silhouette, or (right) the activated R:R state, bound to cyclic guanosine monophosphate (cGMP) and another R-domain shown as a light gray silhouette (PDB ID: 4Z07). The interdomain helices are labeled and shown in red, and G220, G341, and G342 are marked. (**B**) Structural alignment of interdomain helices in the AI and activated conformation. Left: The R-domain in the AI state (solid) and activated state (transparent) conformations are aligned using CNB-A (at right). Middle: orientations of the interdomain helical axes in the AI state (red) and the activated state (black). Right: Cartoon depiction of the interdomain helices only, with a black circle indicating the position of the G220 Cα atom in both the AI state (solid) and the activated state (transparent).

The differences between CNB-B in the AI complex state and the active state (***Kim et al., 2016***) resemble the differences between these states of CNB-A of PKA RI (***Kim et al., 2007***; ***Kim et al., 2005***). The β-sandwich is unchanged while the PKG Iβ PBC is 'open' in the apo, AI state and 'closed' in the activated states (***Figure 5A*** and ***Figure 5—figure supplement 1***). In the AI state, the αB$^B$ helix docks to the C-domain (***Figure 2B***, site 4) in a very different orientation relative to the β-subdomain than is seen in the cGMP-loaded state (***Figure 5B***). N3A$^B$ contacts the open PBC$^B$, filling the space that the αC$^B$ helix occupies in the cGMP-bound state. The αC$^B$ helix, the C-loop, and the capping residue Y351 are disordered, whereas in the activated state the αB$^B$ positions αC$^B$ against PBC$^B$ and Y351 shields cGMP from solvent (***Figure 5A***). Although the orientation of PBC-B with respect to the β-subdomain differs substantially between two states (***Figure 5B***), A309 and L310 at the tip of PBC$^B$ interact with R333 and F336 of the αB$^B$ helix in both states, acting as a hinge.

CNB-A in the auto-inhibited PKG Iβ holoenzyme adopts an 'open' PBC conformation typical of apo CNBs (***Figure 6A,B***), and its contacts with the C-domain indicate that the orientation of the PBC is critical to auto-inhibition. Superimposing with the R:C complex CNB-A shows that L184 and Y188 in the PBC-A helix would clash with M579 and Y583 of the C-domain αG helix, so the 'closed' state is sterically incompatible with binding to the C-domain (***Figure 6C***). In the context of an R:R dimer, however, the 'closed' PBC of cGMP-bound CNB-A contacts the αB helix of CNB-A in another monomer (***Figure 6D***; ***Kim et al., 2016***). Thus, cGMP binding stabilizes a closed CNB-A PBC that

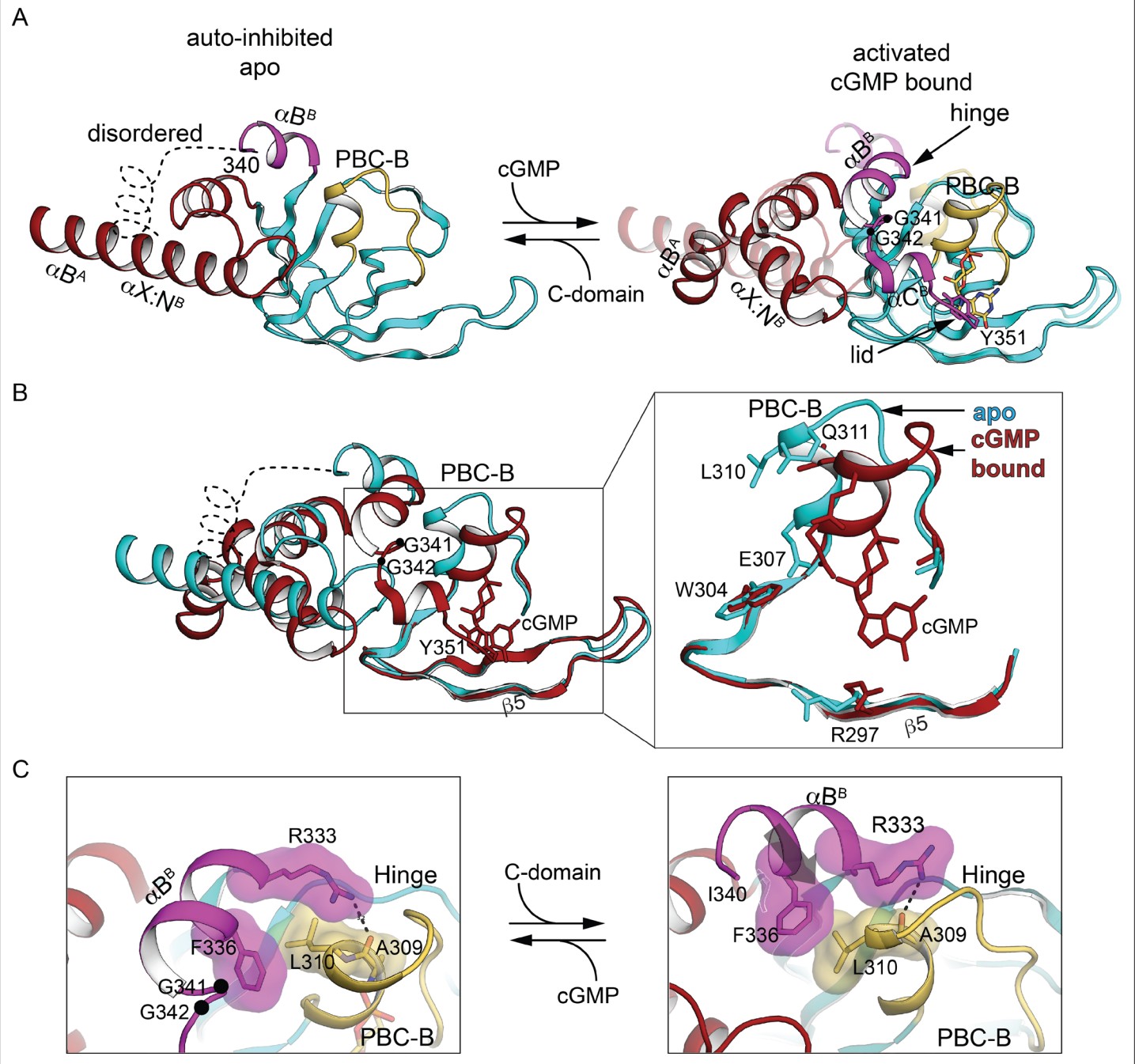

**Figure 5.** The CNB-B helical subdomains are structurally distinct in AI and activated conformation. (**A**) Overall structure of the interdomain helices and CNB-B in the AI (left, PDB ID: 7LV3) and superposition of the cyclic guanosine monophosphate (cGMP)-bound, activated state of the tandem CNB dimer (solid) and AI (transparent) (right, PDB ID: 7LV3 and 4Z07). Hinge glycine Cα atoms are labeled and cGMPs are shown in stick. In the cGMP-bound 'closed' state, PBC-B moves along with αBᴮ helix and the hydrogen bond interactions between αC helix (lid) and PBC-B provide cGMP capping interactions. (**B**) Superposition of CNB-B in the tandem CNB cGMP-bound dimer (red, PDB ID: 4Z07) and in the AI complex (cyan, PDB ID: 7LV3) using the invariant β domain elements. Left: β4, β5, and the β4–β5 loop superimpose well (bottom) but the interdomain helices adopt very different conformations. Right: Zoomed-in view of the differences between the phosphate-binding cassette (PBC) loops of the two states. In the AI 'open' state, the αB helix of CNB-B docks to the C-domain, the αC helix is disordered, and the more open PBC-B interacts directly with N3A. (**C**) Zoomed-in view of the hinge region. Key hinge residues are shown in stick with transparent surface.

The online version of this article includes the following figure supplement(s) for figure 5:

**Figure supplement 1.** Superposition of the PKG Iβ CNB-B domain in the auto-inhibited state (teal, PDB ID: 7LV3) with the isolated CNB-B domain apo state (red, PDB ID: 4KU8) and cyclic guanosine monophosphate (cGMP)-bound state (gray, PDB ID: 4Z07).

**Figure supplement 2.** A conformation induced by cyclic guanosine monophosphate (cGMP) is prevented by $R_P$-cGMPS.

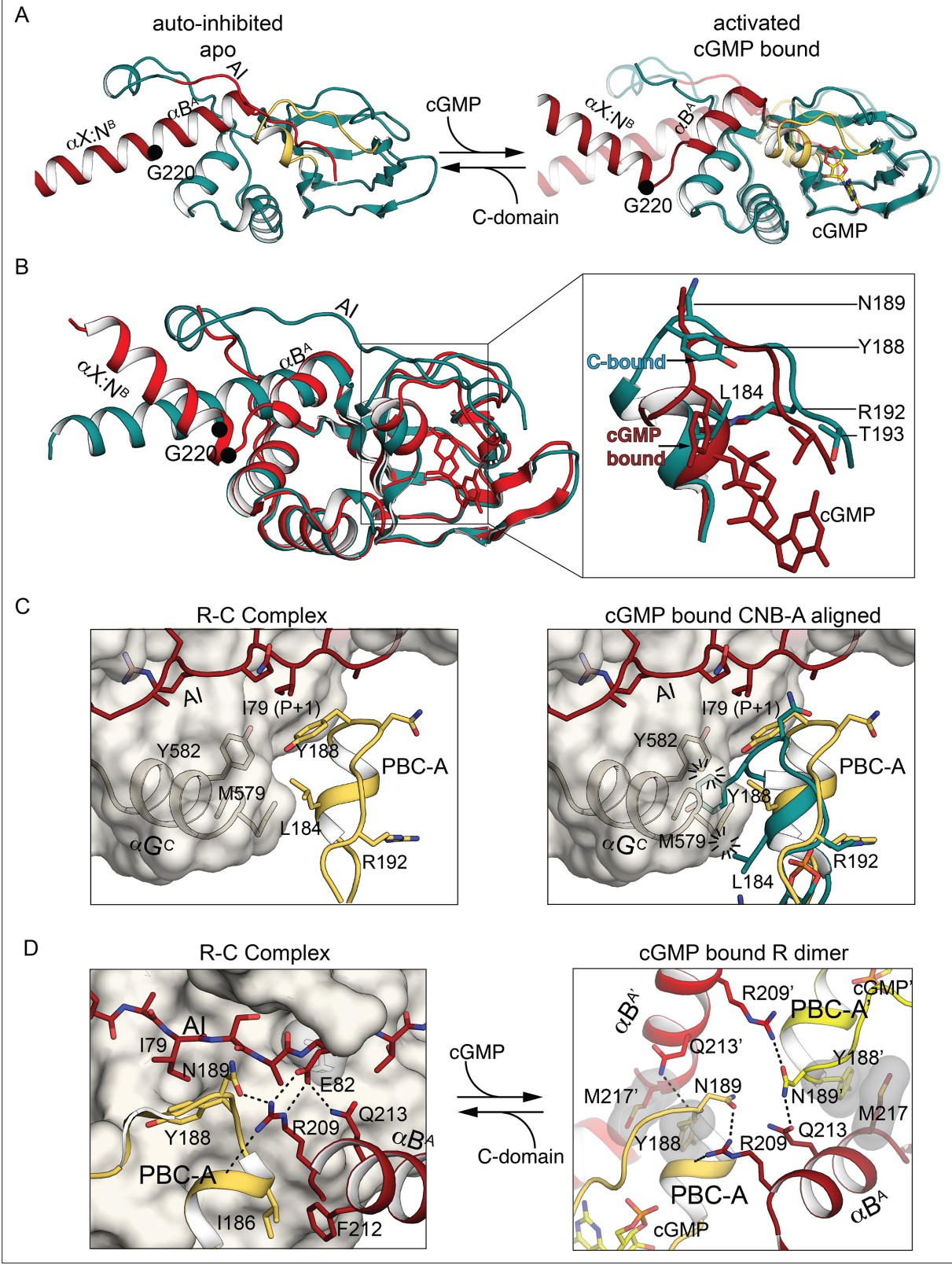

**Figure 6.** CNB-A phosphate-binding cassette (PBC) and interdomain linker differences in auto-inhibition compared to activation. (**A**) Overall structure of the interdomain helices and CNB-A in the auto-inhibited (left, PDB ID: 7LV3) and superposition of the cyclic guanosine monophosphate (cGMP)-bound, activated state of the tandem CNB dimer (solid) and auto-inhibited (transparent) (right, PDB ID: 7LV3 and 4Z07). The same color scheme is used as in *Figure 2*. The Cα atom of G220 is marked with a sphere. (**B**) Structural alignment of cGMP- (PDB ID: 4Z07) and C-domain-bound conformation (PDB ID:

*Figure 6 continued on next page*

*Figure 6 continued*

7LV3). CNB-A in cGMP- and C-bound conformations are colored in red and dark teal, respectively. Zoomed-in view at the right panel shows PBC and αB helix. Key hinge residues and R:C interface residues are shown in stick. (**C**) Alignment of CNB-A:cGMP with PKG holo complex. Left: R:C interface near the apo PBC-A is shown. PBC-A and the αG helix are shown in cartoon. The C-domain is shown as transparent surface. Key R:C interface residues are shown in sticks. Right: Structural alignment between CNB-A:cGMP with PKG Iβ holoenzyme. The residues L184 and Y188 of PBC-A of the cGMP-bound state show steric clashes with M579 and Y583 of the αG helix at the C-domain and suggest that CNB-A:cGMP is not compatible with the C-domain. (**D**) Dynamic AI region replaces the R:R contacts upon the R:C complex formation. Left: The R:R dimer interface near PBC-A is shown. Residues from the second chain are marked with '. Right panel shows AI, PBC-B, and B helix of the R-domain in cartoon and the C-domain in surface. R209 helps assemble a docking surface consisting of AI, PBC-B, and B helix of the R-domain.

The online version of this article includes the following figure supplement(s) for figure 6:

**Figure supplement 1.** Structural basis of the constitutive activation in Thoracic Aortic Aneurysms and Dissections (TAAD) causing mutant.

**Figure supplement 2.** Molecular dynamics (MD) simulations.

**Figure supplement 3.** Representative $^{1}$H,$^{15}$N HSQC cross-peaks of the apo wt (black), cyclic guanosine monophosphate (cGMP)-bound wt (red) and apo R192Q (purple) PKG1β (92–227) (**A**, **B** and **C**).

cannot support R:C interactions but does support R:R interactions. Some of the residues at the R:C interface make alternate interactions in the activated R:R dimer (*Figure 6D*). In the auto-inhibited state, Y188 (PBC-A) and Q213 (αB-helix) interact with I79 and P83 of the AI, and M217 (αB-helix) docks to the activation loop residue W531, but in the cGMP-bound activated R:R dimer, Y188 and Q213 from one chain interact with M217 and N189 from the other chain (*Figure 6D*).

In the auto-inhibited state, I186 of the open PBC lies in a hydrophobic core formed by F212 of the CNB-A αB helix and F118, M119, and V155 (*Figure 2B*), but in the activated state I186 is displaced from this core and these contacts are lost in the closed PBC. The PBC undergoes a substantial but very localized structural change between these two states that reorganizes the tip of the PBC: the Cα atoms of residues 185 through 189 are displaced by 3.4–7.5 Å (5.5 Å RMSD), and the Y188 side chain is displaced by up to 8.3 Å, but flanking residues such as R192 are only minimally displaced (*Figure 6B*; *Berman et al., 2005*; *Kim and Sharma, 2021*; *Taylor et al., 2008*).

## Structural basis for a PKG I activating mutation

Mutation R177Q in PKG Iα is associated with TAAD and constitutively activates the holoenzyme despite weakening cGMP affinity for CNB-A more than $10^5$-fold (*Guo et al., 2013*). Introducing the homologous R192Q into PKG Iβ is also activating (*Chan et al., 2020*). In the auto-inhibited state of PKG Iβ, the R192 guanidinium group donates hydrogen bonds to the L154 and G182 carbonyl oxygens; in the isolated cGMP-bound CNB-B domain (*Kim et al., 2011*), it hydrogen bonds to these oxygens and also to a nonbridging oxygen of the cyclic phosphate in cGMP (*Figure 6—figure supplement 1A*). Loss of this contact explains the decreased cGMP affinity, but it is not obvious why the intact enzyme should be activated. Since CNBs canonically adopt 'open' conformations in the apo state and 'closed' conformations with bound cyclic nucleotide, loss of the arginine could cause activation if its contacts with L154 and G182 stabilize the open state more than the closed state.

We determined the structure of the RQ variant of CNB-A (PDB: 7MBJ), revealing that it adopts a closed conformation that mimics the cGMP-bound wildtype CNB-A (*Figure 6—figure supplement 1B,C*). Since the RQ mutation does not alter folding or prevent adoption of the closed conformation, we propose that the CNB-A sequence is biased to adopt the closed state. Such bias would explain why CNB-A binds cGMP 18-fold tighter than CNB-B despite largely conserved contacts (*Kim and Sharma, 2021*): CNB-B samples the open state more strongly, so more free energy of binding is needed to drive the equilibrium to the closed state. Skewing of the equilibrium between open and closed states has also been proposed to explain the 3-order of magnitude difference in cAMP affinities for PKA RIα and the CNB domain of HCN2 (*Moleschi et al., 2015*).

Because the PBC of the RQ CNB-A domain makes intermonomer crystal contacts, it is possible that these contacts help stabilize the closed state. To determine if the closed state would persist without crystal contacts, we performed molecular dynamics (MD) simulations of the fully solvated domain without nucleotide, and of the wildtype domain with or without nucleotide (*Supplementary file 4*). MD trajectories show that all three domains sample conformational space that is more similar to the closed state (cGMP-bound wildtype CNB-A, PDB 3OD0) than to the open state (CNB-A in the auto-inhibited structure) (*Figure 6—figure supplement 2* and *Supplementary file 4*; *VanSchouwen*

*et al., 2015a*). We conclude that neither crystal contacts nor interactions with cGMP are needed to drive either the wildtype or mutant domain to the closed state, supporting our hypothesis that PBC-A is biased toward the closed state.

Binding of cyclic nucleotide induces backbone amide Nuclear Magnetic Resonance (NMR) chemical shift perturbations in CNB domains that reflect the degree to which the closed state is populated (*Akimoto et al., 2015*; *Byun et al., 2020*). To assess the degree to which the isolated R192Q CNB-A domain samples the closed state in solution, we acquired NMR spectra of CNB-A in the apo and cGMP-loaded states, and of the apo R192Q (*Figure 6—figure supplement 3*). Chemical shifts of apo R192Q NH resonances are more similar to those of the cGMP-bound wildtype than to the apo wildtype (*Figure 6—figure supplement 3*), consistent with the mutation inducing a similar conformational change as cGMP binding to wildtype. We therefore conclude that the RQ mutation causes constitutive kinase activation by allowing CNB-A to adopt a closed PBC-A similar to that of cGMP-bound wildtype without binding cGMP.

### PKG Iβ is auto-inhibited in cis

While the structure in the crystal likely represents a domain-swapped dimer (*Gronenborn, 2009*) undergoing auto-inhibition in trans, previous biochemical studies of truncated PKG lacking the LZ dimerization domain have all been consistent with inhibition in cis. PKG Iβ lacking the LZ domain elutes from gel filtration columns as a monomer (*Richie-Jannetta et al., 2003*), and a monomeric PKG Iβ lacking the first 55 residues has low basal activity in the absence of cGMP, showing that the monomer is capable of auto-inhibition (*Kim et al., 2016*). Because auto-inhibition of the native dimeric enzyme might occur in cis or in trans, we used engineered full-length PKG Iβ heterodimers to assess any roles of interchain communication in regulating PKG activity (*Chan et al., 2020*). In this method, cells are cotransfected with an untagged 'active' PKG and with a Flag-tagged 'dead' PKG bearing a

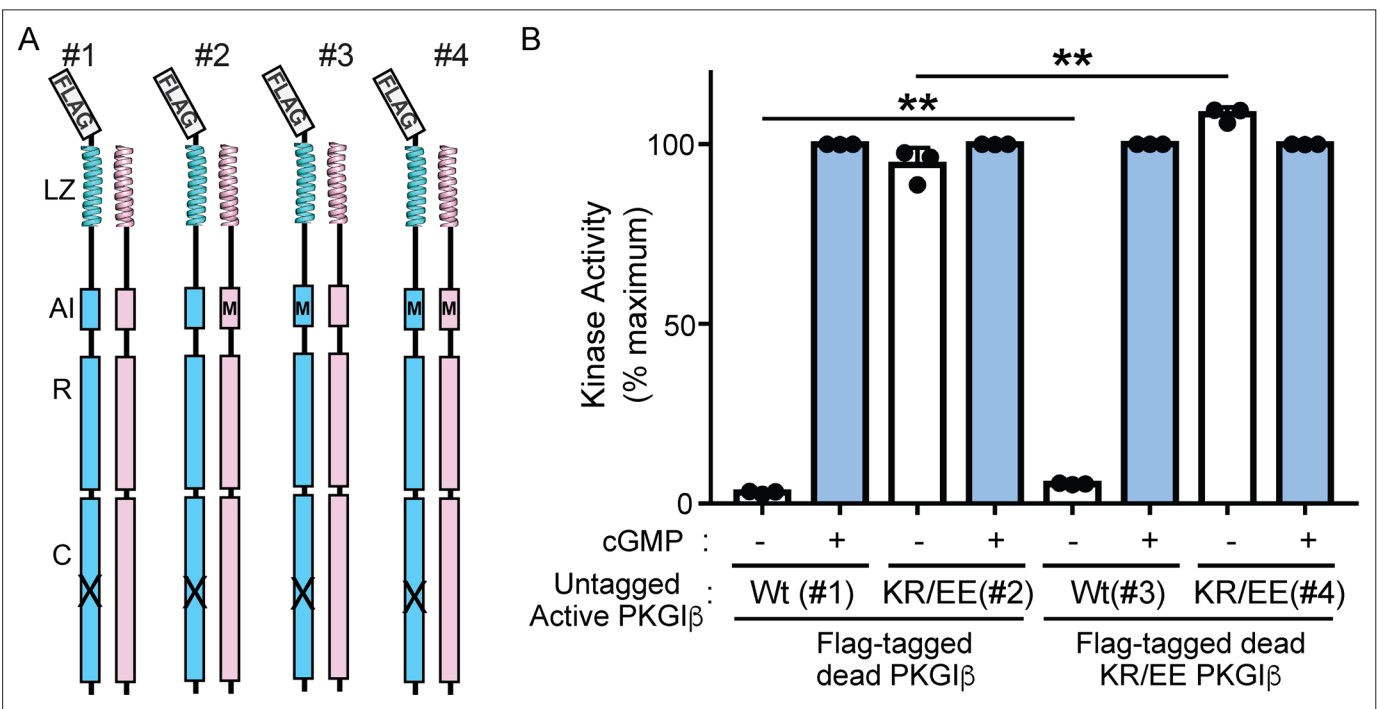

**Figure 7.** The PKG Iβ AI sequence does not contribute to activation through interchain contributions. (**A**) Anti-Flag pulldown of a Flag-tagged (blue) catalytically 'dead' PKG Iβ (X) permits isolation of heterodimers containing an untagged 'active' PKG Iβ (pink) from cotransfected cells. Flag-tagged homodimers that are also present have no kinase activity. Introducing AI mutations K75E and R76E (KR/EE in the text, shown schematically as M) in the 'dead' and/or 'active' chain allows us to distinguish intrachain and interchain influences on auto-inhibition. (**B**) Kinase activity for four heterodimers at zero cGMP (−) or 10 μM cGMP (+) shows that AI mutations in the active monomer (heterodimers 2 and 4) confer constitutive activity and eliminate auto-inhibition. The presence of the AI mutation in the 'dead' monomer alone (heterodimer 3) does not eliminate auto-inhibition, though it may slightly enhance basal activity. Bars indicate mean ± standard deviation from three independent experiments. **p < 0.01, for comparisons between the indicated groups using unpaired two-tailed Student's *t*-test.

point mutation that renders the kinase domain catalytically inactive. When protein is isolated with anti-Flag agarose and eluted with Flag peptide, the resulting purified kinase is a mixture of dead–dead homodimers and dead–active heterodimers; in assays, activity comes only from the heterodimers. In this system, introducing mutations in the dead chain cannot affect the kinase activity of that chain (intrachain effect), since the active site is mutated, so any changes in activity must result from inter-chain effects. We prepared constructs that mutated K75 and R76 to glutamate (KR/EE), reasoning that this would destabilize the auto-inhibited R:C state through charge repulsion at the active site.

We performed kinase assays under basal conditions (no cGMP) and activating conditions (10 μM cGMP) using Flag-tag purified protein from four separate cotransfections that pair Flag-tagged kinase-dead PKG Iβ (with or without the KR/EE mutation) with untagged kinase-active PKG Iβ (with or without the KR/EE mutation) (*Figure 7*). With wildtype active chain, low levels of activity are seen in the absence of cGMP, indicating auto-inhibition, and high levels of activity are seen with cGMP, consistent with activation. When the KR/EE mutation is present on the active chain, similar high activity levels are seen with or without cGMP, indicating that little or no auto-inhibition occurs. We infer that dimeric PKG Iβ undergoes auto-inhibition primarily in cis. The activity of the KR/EE mutant active chain is slightly lower when paired with a wildtype Flag-tagged partner than when paired with the mutant, indicating that the wildtype AI may provide some modest inhibition in trans. The basal kinase activity of the untagged wildtype chain is slightly higher when it is associated with the dead chain bearing the KR/EE mutation (5.4% of maximum) rather than the wildtype AI (3.1% of maximum), suggesting that the presence of a second copy of the AI in trans may modestly enhance auto-inhibition of the heterodimer.

## PKG Iβ activation is potentiated by substrate

Our structure indicates that mutations such as S80A (*Busch et al., 2002*) or KR/EE promote cGMP-mediated activation by weakening R:C interactions between the AI sequence and the active site cleft, and thus destabilizing the auto-inhibited state. Interestingly, the PKG Iβ cGMP activation constant can be modulated by the peptide substrate: PKG Iβ activates at twofold lower cGMP levels with VASP-tide as a substrate than with Kemptide (*Figure 1*, *Figure 1—figure supplement 3*), consistent with a competition between the AI pseudo-substrate and the actual substrate for accessing the catalytic cleft, with binding of substrate preventing formation of the auto-inhibited state. The twofold lower $K_M$ for VASPtide compared to Kemptide (*Figure 1—figure supplement 3*) explains enhanced activation by the former compared to the latter. Competition of substrate with the AI sequence for binding to the catalytic domain has also been invoked in explaining the activation of PKA (*Byun et al., 2020*).

In comparison with PKG Iβ, PKG Iα exhibits higher fractional basal activity, a lower activation constant, and only small differences in cGMP activation constants with Kemptide and VASPtide (*Figure 1D* and *Figure 1—figure supplement 3*). Such differences may reflect not only poorer contacts between the Iα AI sequence and the active site (as discussed above for site 1 of *Figure 2B*), but also more limited access of the Iα AI to the active site compared to Iβ. The Iα-specific LZ, linker, and hinge regions that contribute to cGMP-mediated activation (*Ruth et al., 1997*) might combine to sterically restrict (relative to PKG Iβ) how AI diffusion samples its environment to access the kinase active site cleft. The construct crystallized here lacks the LZ and linker, so structures of full-length proteins may be needed to clarify these isoform-specific aspects of the activation mechanism.

## Discussion
### PBC conformations control access to the auto-inhibited state

In the auto-inhibited PKG Iβ 71–686 structure presented here, the PBC loop of CNB-B and its adjacent helix adopt conformations very similar to those of the isolated apo CNB-B domain; this arrangement allows αA- and αB helices to contact the C-domain, whereas the repositioning of these helices as in the cGMP-bound closed state would disrupt these contacts (*Figures 2B and 5*), as we had previously proposed based on structures of the isolated CNB-B domain (*Figure 5—figure supplement 2*; *Campbell et al., 2017*; *Huang et al., 2014a*; *Huang et al., 2014b*). The open state of CNB-A also supports favorable R:C contacts in auto-inhibited PKG Iβ that would be incompatible with the closed state that is stabilized by binding cGMP (*Figure 4*). The auto-inhibited structure presented here is the first instance in which a fully open state of PKG I CNB-A has been observed.

NMR data show that the isolated apo CNB-B domain is predominantly open but can sample the closed state (*Huang et al., 2014b*; *VanSchouwen et al., 2015b*). The closed state is stabilized by cGMP binding, while the PKG Iβ 71–686 structure provides a snapshot of the contacts that stabilize the open state of apo CNB-B and support auto-inhibition. If the isolated CNB-A domain also samples the open and closed states, the tighter cGMP affinity of CNB-A compared to CNB-B (*Kim et al., 2011*) could be explained by apo CNB-A being biased to the closed state whereas CNB-B is biased to the open state. Similar coupling of cyclic nucleotide binding to differently skewed equilibria between the open and closed states has been proposed to explain differences in cAMP affinities for PKA RIα and the CNB domain of HCN2 (*Moleschi et al., 2015*). We propose that the open CNB-A domain seen in the auto-inhibited enzyme is a high-energy state stabilized by R:C contacts that rapidly reverts to the closed state when these contacts are lost. The auto-inhibited structure provides a basis for understanding how two analogous CNB domain switches with very different intrinsic setpoints are coupled to one another in an allosteric mechanism that exhibits one macroscopic activation constant.

CNB-B discriminates strongly between cGMP and cAMP (*Huang et al., 2014b*), and certain contacts are critical to confer the conformational change, as highlighted by the structure of the isolated CNB-B domain in an inhibitor-bound open state (*Campbell et al., 2017*; *Huang et al., 2014a*; *Huang et al., 2014b*): although $R_P$-cGMPS makes many of the same contacts as cGMP, the PBC loop does not close to contact the phosphorothioate sulfur, and the adjacent helix does not reposition (*Figure 5—figure supplement 2*). Whereas cGMP binding to PKG Iβ CNB-B disrupts its αA- and αB-helix contacts with the C-domain, resulting in activation, we expect that $R_P$-cGMPS binding will not disrupt the R:C interface of the auto-inhibited state, explaining why this cGMP analog acts as an inhibitor rather than an activator.

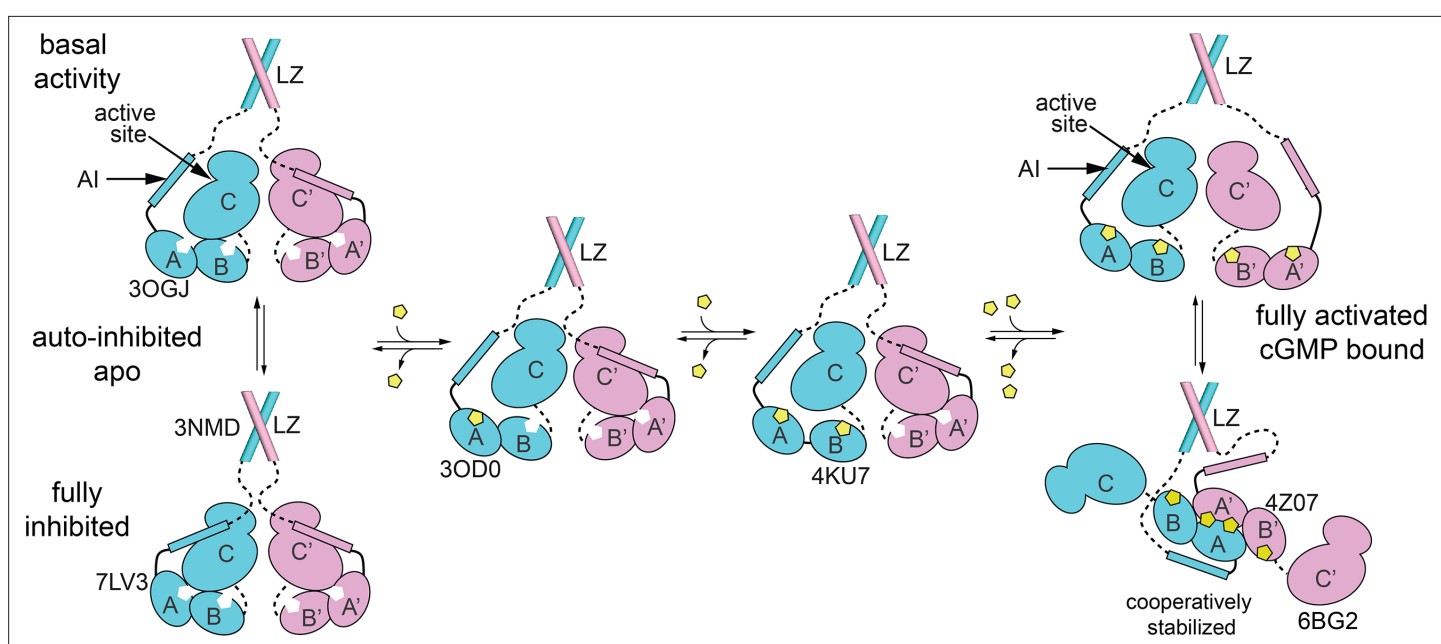

**Figure 8.** Schematic model for how equilibria between different states of PKG Iβ support auto-inhibition and activation. Crystal structures that represent one or more domains in each state are indicated by PDB code. The auto-inhibited state (bottom left) dimerizes through leucine zipper domains (3NMD) that keep the cyclic guanosine monophosphate (cGMP)-free, auto-inhibited catalytic domains (7LV3) in close proximity. The AI sequence of cGMP-free PKG Iβ can transiently leave the active site and permit basal activity (top left). Binding of cGMP (yellow pentagon) to CNB-A (middle left; 3OD0) and/ or CNB-B (middle right, 4KU7) lock the CNBs into conformations that are not compatible with the auto-inhibited state. Levels of cGMP that saturate CNBs of both monomers enable complete activation (top right) and support formation of a complex between the tandem CNB domains that provides cooperativity and may restrict the ability of the AI sequence to access the catalytic active site. The discussion further describes how peptide substrate and cGMP analogs, including cAMP, can influence the equilibria.

The online version of this article includes the following figure supplement(s) for figure 8:

**Figure supplement 1.** Solvent accessibility of cyclic guanosine monophosphate (cGMP)-binding pockets.

## Activation mechanism and cooperativity

We propose a model for PKG Iβ activation that combines available biochemical data to posit roles for several distinct enzyme substates that we relate to known crystal structures (*Figure 8*). At low cGMP concentration, the AI sequence and the helical sub-domains of the R-domain dock to the C-domain as in our auto-inhibited complex, blocking the active site and shutting off kinase activity (*Figure 8*, bottom left). Our mutational analysis with full-length heterodimers (*Figure 7*) establishes that this auto-inhibition occurs within each PKG Iβ monomer; this may not hold for all PKG isoforms.

Basal kinase activity in the absence of cGMP occurs during transient excursions of the AI away from the active site (*Figure 8*, top left). AI disorder and exposure to solvent during these excursions explain the complete hydrogen/deuterium exchange of AI backbone amides in the wildtype apo PKG Iβ (*Chan et al., 2020*). The frequency of these excursions should depend on the AI sequence and phosphorylation state, which influence the detailed interactions between the AI and the C-domain as discussed above. Phosphorylation of S80 during an excursion would strongly bias against a return to the auto-inhibited state, and occupation of the active site by a competitor would prolong an excursion. Since PKG Iβ shows about 40-fold activation (*Figure 1D* and *Figure 1—figure supplement 3*), in the absence of cGMP we expect that transient excursions expose the kinase active site about 2.5% of the time.

The cGMP sites of both CNBs are accessible to solvent in the auto-inhibited state (*Figure 8— figure supplement 1*), so cGMP could bind to either of these when its intracellular concentration rises; first binding to the high affinity A site may be more likely (*Figure 8*, middle left). Binding of cGMP to CNB-A causes a conformational change that reorients PBC-A, forces CNB-A away from the C-domain, and enhances the tendency of the AI to leave the active site, consistent with the partial activation associated with cGMP binding to the A site (*Corbin and Døskeland, 1983*).

Binding of a second cGMP to the lower affinity, highly selective B site (*Figure 8*, middle right) accompanies reorganization of the helical subdomain as seen in the isolated CNB-B bound to cGMP (*Figure 5*; *Huang et al., 2014b*), weakens R:C interactions at sites 3 and 4, and releases the CNB-B αA and αB helices from the C-domain. With neither CNB bound to the C-domain, the domains diffuse somewhat independently and the interdomain linker can sample nonhelical conformations consistent with small-angle scattering data for monomeric PKG Iβ showing that cGMP binding to both sites is required for R:C dissociation, elongation, and full activation (*Wall et al., 2003*). The auto-inhibited state fixes the relative positions of domains within a monomer (*Figure 8*, bottom left), but in activated states these domains may behave in solution like beads on a string.

PKG Iβ is activated cooperatively by cGMP (Hill coefficient of 1.6) (*Smith et al., 2000*), and the two tandem CNB domains suggest that interactions within one monomer might mediate cooperativity. However, auto-inhibited monomeric PKG Iβ LZ deletion mutants are activated by cGMP without exhibiting cooperativity (*Døskeland et al., 1987*; *Wall et al., 2003*), so dimerization is dispensable for auto-inhibition but essential for cooperative activation. We propose that intermonomer interactions in the activated state provide the basis for cGMP cooperative activation. If each monomer of an LZ-mediated dimer has two bound cGMP, the PBC-A and bent interdomain helices of one monomer can associate with the same regions of the other monomer to which it is tethered and assemble into an antiparallel R:R interface (*Figures 4A and 6D*; *Kim et al., 2016*). The formation of this interface further stabilizes the activated state (*Figure 8*, bottom right). This assembly helps sustain kinase activation since it occupies much of the R-domain surface that would otherwise be free to associate with the C-domain; it also shields the bound cGMP from solvent, which reduces the rate at which it might be released and degraded by phosphodiesterases. Fixing the CNB domain orientations in this complex may also influence access of the AI sequence to the kinase active site. Dynamic assembly and disassembly of the R:R interface and release and rebinding of cGMP would allow the occupancy of the R:R state to be coupled to the instantaneous cGMP concentration; when cGMP levels fall, the R:R state will be depopulated, and the R-domain will be available to interact with and inhibit the C-domain.

In this model of cGMP-mediated PKG activation (*Figure 8*), the rapid, reversible sampling of states is biased away from the auto-inhibited enzyme default by PBC conformational changes from 'open' to 'closed' that are coupled to binding of cGMP. Because cAMP can bind to the CNBs and induce the closed state (*VanSchouwen et al., 2015b*), the model predicts that cAMP will act as a partial agonist for the holoenzyme, especially through binding to CNB-A, which discriminates poorly between cGMP and cAMP (*Kim et al., 2011*; *Lorenz et al., 2017*). Mutations that favor the PBC 'closed' state, or that

favor the conformational changes that occur in response to the 'open' to 'closed' transition, will also favor activation. The activating effect of the CNB-A RQ mutation (*Chan et al., 2020*; *Guo et al., 2013*) reflects destabilization of the auto-inhibited state we describe here.

The model can be expanded by adding states of the holoenzyme, including those with bound inhibitors. The cGMP analog $R_P$-cGMPS is a PKG inhibitor that makes many of the same contacts as cGMP with the isolated CNB-B domain, but the PBC loop does not close to contact the phosphoro-thioate sulfur, and the adjacent helix does not reposition (*Campbell et al., 2017*; *Figure 5—figure supplement 2*). Binding of $R_P$-cGMPS can therefore occupy the CNB-B site, competing with any cGMP that is present without disrupting the CNB-B αA- and αB-helix contacts with the C-domain, explaining why this cGMP analog does not block basal activity but inhibits activation (*Campbell et al., 2017*).

The activation model (*Figure 8*) can explain the substrate dependence of the PKG Iβ cGMP acti-vation constant (*Figure 1D* and *Figure 1—figure supplement 3*) when its states are expanded to include competition between inter- and intramolecular binding at the active site. Substrate binding, which can occur even in the absence of cGMP when the AI sequence makes a transient excursion, pulls the coupled equilibria of *Figure 8* to the right. The degree to which the activation constant shifts should depend on the affinity and concentration of the peptide, which determine the bound fraction in competition with binding by the AI sequence. Substrate-induced shifting of the activation curve rules out models in which substrate binding occurs only after activation; the coupled equilibria imply that every species in *Figure 8* with a free active site cleft can reversibly adopt a corresponding substrate-bound state. Colocalization of PKG Iβ with targets through its isotype-specific N-terminal domain (*Casteel et al., 2005*) may therefore not only influence the choice of substrate but also enhance the degree to which the enzyme is activated.

Although the auto-inhibited state described here (*Figure 8*, bottom left) fixes the relative positions of the CNBs and C-domain within one monomer, in other states the interdomain linkers may allow a monomer to behave more dynamically, like domain beads on a string. The shorter linker and the distinct LZ and hinge regions of PKG Iα may decrease the opportunities for random diffusion to juxta-pose the AI with the active site cleft, resulting in a lower AI effective local concentration at the active cleft for PKG Iα compared to Iβ. This physical rationale would explain why PKG Iα shows minimal substrate dependence in its cGMP activation constant (*Figure 1D* and *Figure 1—figure supplement 3*): substrates will compete better with an AI that more rarely accesses the active site cleft. This ratio-nale may also explain how changes to Iα-specific residues outside the AI impact activation (*Ruth et al., 1997*). Mutation C43S substantially increases the cGMP activation constant for PKG Iα (*Kalyanaraman et al., 2017*), which could be explained by the loss of a C43-mediated disulfide in the LZ (*Qin et al., 2015*) that increases dynamic motions and facilitates AI access to the kinase active site cleft.

This model and our structure provide rationales for how changes in the AI affect PKG Iα and Iβ isoform cGMP activation constants and fractional basal activities by affecting AI contacts with the active site, but they do not provide detailed insights as to how the sequences and structures of the isoform-specific LZs and other elements in the N-terminal domain might affect access to any of the states in the model. The only explicit role for the LZs in the model of *Figure 8* is to promote sampling of the R:R state, whose interacting residues are present in both isoforms; how isoform differences might structurally influence access to this state is not known since the LZs were truncated in the construct crystallized here. Explanations for some of the observed functional differences between the Iα and Iβ isoforms await more detailed models, and perhaps structures of full-length enzyme isoforms in different states of activation or inhibition.

## Materials and methods
### Construct design, protein expression, and purification of PKG Iβ (71–686)

The sequence encoding human PKG Iβ (residues 71–686) was cloned into pBlueBacHis2A vector with tobacco etch virus (TEV) protease site before coding sequence. The protein was expressed in SF9 cells with multiplicity of infection of 2 for 48 hr. Cells were suspended in 200 mM potassium phosphate (pH 7.5), 500 mM NaCl (sodium chloride), 1 mM β-mercaptoethanol (Buffer A) and lysed with a cell disruptor (Constant Systems, Daventry Northants, UK). The supernatant was loaded onto HisTrap HP 5 ml column (GE Healthcare) and eluted with buffer A containing 300 mM imidazole on an ÄKTA

purifier system (GE Healthcare). The His-tag was removed by incubating the eluted protein with TEV protease overnight at 4°C. A second nickel affinity chromatography step was performed to remove TEV protease and flow-through fractions were collected. The tagless protein was further purified by anion exchange chromatography (Mono Q 10/100 GL, GE Healthcare) in 25 mM potassium phosphate (pH 7.5), 1 mM β-mercaptoethanol with and without 1 M NaCl. The Mono Q peak fractions were pooled, concentrated and passed through a Hiload 10/300 Superdex 200 column (GE Healthcare) equilibrated with 25 mM 2-(N-morpholino)ethanesulfonic acid (MES), 150 mM NaCl, 1 mM tris(2-carboxyethyl)phosphine (TCEP), 2 mM $MnCl_2$, and 0.1 mM AMP-PNP.

## Expression and purification of PKG Iα (79–212)

A DNA construct containing His-tagged human PKG Iα (79–212) R177Q was cloned into pQTEV and transformed into TP2000 *E. coli* cells (**Kim et al., 2015**; **Roy and Danchin, 1982**). Cells were grown at 37°C until $OD_{600}$ of 0.6 and induced with 0.5 mM Isopropyl ß-D-1-thiogalactopyranoside (IPTG). Cells were then grown for an additional 10 hr at 25°C, harvested by centrifugation. Cells were suspended in Buffer A (50 mM potassium phosphate, 500 mM NaCl, 1 mM β-mercaptoethanol [pH 7.5]) and lysed with a cell disruptor (Constant Systems, Daventry Northants, UK). The protein was purified with BioRad IMAC resin on a BioRad Profinia purification system and eluted with Buffer A containing 300 mM imidazole. The elution fraction protein sample was incubated with TEV protease at 4°C overnight for His-tag removal. A second nickel affinity chromatography step was performed to remove TEV protease and flow-through fractions were collected. Protein was further purified with gel filtration on a Hi-load 16/60 Superdex-75 column (GE Healthcare) in 25 mM Trizma (pH 7.5), 150 mM NaCl, and 1 mM TCEP–HCl.

## Crystallization, data collection, phasing, model building, and refinement

To obtain crystals, PKG Iβ (71–686) protein was concentrated to 20 mg ml⁻¹ by using 30 kDa cutoff Amicon Ultra (Millipore) and initial crystal screening was performed by Mosquito Crystal robot (TTP Labtech) with 100 nl protein drop and 100 nl crystallization solution over 70 µl well. Further optimization was performed with 250 nl protein and 250 nl crystallization solution. Crystals were obtained by vapor diffusion method in 120 mM ethylene glycol, 100 mM Bicine (pH 8.5), 20 % (wt/vol) PEG 8000 at room temperature. The crystals were flash cooled in liquid nitrogen. Diffraction data were collected on Beamline 5.0.1 (Advanced Light Source, Berkeley, CA, USA). To obtain crystals of PKG Iα (79–212) R177Q, 10 mM cGMP (Aral Biosynthetics) was added to the purified protein, and concentrated to 20 mg ml⁻¹ with a 10 kDa cutoff Amicon Ultra (Millipore). Initial crystal screening was performed by Mosquito Crystal robot (TTP Labtech) with 300 nl protein drop and 300 nl crystallization solution over 70 µl well. Further optimization was performed with 2 µl protein and 2 µl crystallization solution over 500 µl well. Crystals were obtained by vapor diffusion method in 6% Tacsimate (pH 4.5), 18% PEG 3350 at 22°C. The R177Q crystals were dipped in cryoprotectant (Paratone-N) and flash cooled in liquid nitrogen. Diffraction data were collected on Beamline 5.0.1 (Advanced Light Source, Berkeley, CA, USA). Diffraction data for both crystals were processed by using CCP4.iMosflm (**Battye et al., 2011**) and structure determined by Phaser MR using the crystal structure of PKA holoenzyme (PDB ID: 2QCS) and cGMP-bound PKG Iβ CNB-A structure (PDB ID: 4Z07) as molecular replacement probes (**McCoy, 2007**). Coot was used for model building of the structures and Phenix.Refine was used for refinement (**Afonine et al., 2012**; **Emsley and Cowtan, 2004**). All the figures were generated using PyMOL (DeLano Scientific).

## Small-angle X-ray scattering data collection and analysis

PKG Iβ (71–686) protein at 3.5 mg ml⁻¹ at 20°C was exposed for 3 s at ALS beamline 12.3.1 and small-angle X-ray scattering data were collected over the $q$ range 0.01–0.39 Å⁻¹. Primary data reduction and processing were performed with the ScÅtter pipeline and ATSAS suite. The fitted values for $I(0)$ and $R_g$ from Guinier analysis and $P(r)$ analysis agree closely see **Supplementary file 2**; $I(0)$ is consistent with a monomer and the experimental $R_g$ is consistent with the cis-conformation. Ab initio analysis was performed with DAMMIF/DAMMIN, computation of model intensities was performed with the FoXS webserver, and crystal structures were fit into models using SUPCOMB.

## Cloning, protein expression, and purification of full-length PKG Iα and Iβ

FLAG-TwinStrep-tagged hPKG Iα and Iβ were generated by cloning the respective DNA sequence from pFLAG hPKG Iα and pFLAG hPKG Iβ (*Kalyanaraman et al., 2017*) using 5′-AAA GCT AGC AGC GAG CTA GAG GAA GAC TTT GCC-3′ and 5′-AAA CTC GAG TTA TTA GAA GTC TAT ATC CCA TCC T GA-3′ as primers for PKG Iα and 5′- GAA GCT AGC ATC CGG GAT TTA CAG TAC G-3′ and 5′- AAA CTC GAG TTA TTA GAA GTC TAT ATC CCA TCC TGA-3′ for PKG Iβ. Both PCR products were subcloned by *Nhe*I and *Xho*I digestion and subsequent ligation into the *Nhe*I and *Xho*I digested pcDNA 3.0 SF TAP plasmid (*Gloeckner et al., 2007*). The generated plasmids were analyzed by Sanger sequencing. Both constructs were expressed in HEK293-T cells grown to a confluency of ~80% and transfected with the respective plasmid using polyethylenimine. Cells were harvested and freeze thawed for purification. Subsequent cell lysis was performed in lysis buffer containing 50 mM Tris (pH 7.4), 150 mM NaCl, 0.5 mM TCEP, protease inhibitor cocktail (Roche) phosphatase inhibitor (Roche) and 0.4% Tween-20. Proteins were purified using Strep-Tactin Superflow resin (IBA Lifesciences) employing four washing steps. First, lysis buffer, followed by a high phosphate buffer (366 mM $Na_2HPO_4$, 134 mM $NaH_2PO_4$, 0.5 mM TCEP) and two times Tris buffer (50 mM Tris, 150 mM NaCl, 0.5 mM TCEP). Elution was performed using a Tris buffer containing 2.5 mM desthiobiotin. Subsequently, a buffer exchange with Tris buffer was performed to store the proteins without the desthiobiotin to avoid possible interference with downstream analyses.

## PKG Iα and Iβ in vitro kinase assays

Specific kinase activities, activation constants ($K_{act}$) and Michaelis–Menten constants ($K_M$) were determined using an enzyme-coupled spectrophotometric kinase assay as described by *Cook et al., 1982*. For specific kinase activity, assay buffer (100 mM 3-(N-morpholino)propanesulfonic acid(MOPS), 10 mM $MgCl_2$, 1 mM ATP, 1 mM phosphoenolpyruvate, 15.1 U/ml lactate dehydrogenase, 8.4 U/ml pyruvate kinase, 230 µM 1,4-Dihydronicotinamide adenine dinucleotide (NADH), 5 mM β-mercaptoethanol) was mixed with the hPKG samples in a glass cuvette and the reaction was started by adding the respective substrate peptide (VASPtide: RRKVSKQE; Kemptide: LRRASLG) to a final concentration of 1 mM. For basal activities, measurements were performed without cGMP. Maximal kinase activity was determined in the presence of a final cGMP concentration of 200 µM. The absorption at 340 nm was monitored for 30–120 s using a double beam photometer (Specord 205, Analytik Jena) and the slope was determined to calculate the specific kinase activity.

The activation constant ($K_{act}$) was determined by mixing assay mix with a kinase mix and a dilution series of cGMP in a 384-well plate (Microplate, 384 well, PS, F-bottom, clear, Greiner Bio-One) in a ratio of 2:1:1 and monitoring the absorption at 340 nm for 200–300 s in a microplate reader (CLARIOstar, BMG Labtech). Protein activity was calculated according to the Lambert–Beer law and converted to the observed catalytic activity at the given substrate concentration of 1 mM ($k_{obs}$). Values of $k_{obs}$ were plotted against the cGMP concentration on a logarithmic scale and $K_{act}$ was determined by fitting the data using a sigmoidal dose–response fit with a variable slope. The activation constant is defined as the cGMP concentration at which half-maximal kinase activity occurs.

A similar procedure was followed for the determination of Michaelis–Menten constants ($K_M$). Here, assay mix, kinase mix with cGMP (final: 50 µM cGMP) and a dilution of VASPtide or Kemptide, respectively, were mixed in a 384-well microplate in a 2:1:1 ratio and processed as described. Calculated kinase activities were plotted against the substrate concentration and fitted to the Michaelis–Menten equation to obtain $K_M$ and $v_{max}$.

## PKG Iβ heterodimer in vitro kinase assays

The PKG Iβ complexes were purified from 293T cells cotransfected with Flag-tagged dead and untagged active PKG Iβ expression plasmids, as described (*Chan et al., 2020*). Purified proteins were diluted in KPEM Buffer (10 mM potassium phosphate [pH 7.0], 1 mM 2,2′,2′′,2′′′-(Ethane-1,2-diyldinitrilo) tetraacetic acid (EDTA), and 25 mM mercaptoethanol) and reactions were initiated by adding to reaction mix with or without 10 µM cGMP. Final reactions contained: 40 mM 2-[4-(2-hydroxyethyl) piperazin-1-yl]ethanesulfonic acid (HEPES) (pH 7.0), 8 µg Kemptide, 10 mM $MgCl_2$, 60 µM ATP, and 0.6 µCi $^{32}P$-γ-ATP. Reactions were incubated at 30°C for 1.5 min. and stopped by spotting on P81

phosphocellulose paper. Unincorporated $^{32}$P-γ-ATP was removed by washing in four times 2 l of 0.452% *o*-phosphoric acid, and $^{32}$PO$_4$ incorporation was measured by liquid scintillation counting.

## PKG Iβ CNB-A domain NMR

The PKG Iβ 92–227 construct was expressed with an N-terminal His tag, in BL21(DE3) *E. coli* cells grown at 37°C in $^{15}$N M9 media. Expression was induced with 0.5 mM IPTG at an OD$_{600}$ of 0.8, and the cells were grown for 16 hr at 18°C. PKG Iβ 92–227 was purified as reported (*VanSchouwen et al., 2015b*). NMR samples were prepared in 50 mM Tris, pH 7.0, 100 mM NaCl, 1 mM 1,4-dithiothreitol (DTT), 0.2% (wt/vol) NaN$_3$. An apo sample was prepared by concentrating the purified PKG to 100 μM, and adding 5% (vol/vol) D$_2$O, phosphodiesterase (PDE), 1 mM ATP, and 10 mM MgCl$_2$. A cGMP-bound sample was prepared similarly, with the addition of 1 mM cGMP to the apo protein solution. All HSQC spectra were recorded with 16 scans and 1-s recycle delay. The spectra included 128 (t1) and 2048 (t2) complex points with spectral widths of 38.0 and 16.2 ppm for the $^{15}$N and $^1$H dimensions, respectively. Carrier frequencies of $^1$H and $^{15}$N were set at the water and central amide region, respectively. All spectra were acquired at 300 K with a Bruker Avance 700 MHz NMR spectrometer equipped with a 5 mm TCI cryoprobe. The spectra were processed with TOPSPIN and analyzed using Sparky (*Lee et al., 2015*).

## MD simulation protocol

All MD simulations were performed using the NAMD 2.12 software (*Phillips et al., 2005*) on the Shared Hierarchical Academic Research Computing Network (SHARCNET), following a previously described protocol (*VanSchouwen and Melacini, 2018*), with solvent box dimensions of 49 Å for the wildtype CNB-A domain structures, 50 Å for the RQ-mutant CNB-A domain structure, or 186 Å for the full-length cGMP-free dimer structure. All simulations were executed using 2.1 GHz 32-core Broadwell compute nodes accelerated with two NVIDIA Pascal GPUs per node. The CNB-A domain simulations were each executed for >500 ns at constant temperature and pressure, while the full-length cGMP-free dimer simulation was executed in triplicate for 200 ns at constant temperature and pressure, saving structures every 100,000 timesteps (i.e. 100.0 ps) for subsequent analysis. A summary of the MD simulations is given in *Supplementary file 4*.

## Acknowledgements

We thank Kim lab members for critical reading of the manuscript. C.K. was funded by NIH grant R01 GM090161. The BCM Macromolecular X-ray Crystallography Core is supported in part by an NIH Shared Instrumentation Grant Award (S10OD030246). G.M. was Canadian Institutes of Health Research grant 389522. F.W.H. was supported by a grant of the Deutsche Forschungsgemeinschaft (He1818/10-1), the Kassel Graduate school 'clocks', and Federal Ministry of Education and Research, Germany (TargetRD, FKZ: 16GW0270 to F.W.H.). The Berkeley Center for Structural Biology is supported in part by the NIH, the National Institute of General Medical Sciences, and the Howard Hughes Medical Institute. The Advanced Light Source is supported by the Director, Office of Science, Office of Basic Energy Sciences, of the U.S. Department of Energy under contract no. DE-AC02-05CH11231. The SIBYLS beamline (ALS) is supported in part by US DOE program Integrated Diffraction Analysis Technologies and NIH project ALS-ENABLE (P30 GM124169) and a High-End Instrumentation Grant S10OD018483.

## Additional information

### Funding

| Funder | Grant reference number | Author |
| --- | --- | --- |
| National Institute of General Medical Sciences | R01 GM090161 | Choel Kim |
| Deutsche Forschungsgemeinschaft | He1818/10-1 | Friedrich W Herberg |

| Funder | Grant reference number | Author |
|---|---|---|
| Federal Ministry of Education and Research, Germany | TargetRD | Friedrich W Herberg |
| Federal Ministry of Education and Research | FKZ: 16GW0270 | Friedrich W Herberg |
| Canadian Institutes of Health Research | Research grant 389522 | Giuseppe Melacini |

The funders had no role in study design, data collection, and interpretation, or the decision to submit the work for publication.

## Author contributions

Rajesh Sharma, Data curation, Validation, Methodology, Writing – review and editing; Jeong Joo Kim, Data curation, Formal analysis, Methodology; Liying Qin, Data curation, Methodology; Philipp Henning, Data curation, Formal analysis, Methodology, Writing – review and editing; Madoka Akimoto, Gundeep Kaur, Data curation, Investigation, Writing – review and editing; Bryan VanSchouwen, Data curation, Formal analysis, Investigation, Methodology, Writing – review and editing; Banumathi Sankaran, Data curation, Methodology, Writing – review and editing; Kevin R MacKenzie, Conceptualization, Formal analysis, Validation, Investigation, Visualization, Writing – original draft, Writing – review and editing; Giuseppe Melacini, Resources, Data curation, Formal analysis, Supervision, Investigation, Visualization, Project administration, Writing – review and editing; Darren E Casteel, Conceptualization, Data curation, Formal analysis, Validation, Investigation, Visualization, Writing – original draft, Writing – review and editing; Friedrich W Herberg, Resources, Formal analysis, Supervision, Funding acquisition, Validation, Investigation, Visualization, Methodology, Writing – original draft, Writing – review and editing; Choel Kim, Conceptualization, Resources, Data curation, Formal analysis, Supervision, Funding acquisition, Validation, Investigation, Visualization, Methodology, Writing – original draft, Project administration, Writing – review and editing

## Author ORCIDs

Kevin R MacKenzie http://orcid.org/0000-0001-9620-5868
Darren E Casteel http://orcid.org/0000-0002-7673-6597
Choel Kim http://orcid.org/0000-0002-3152-0020

## Decision letter and Author response

Decision letter https://doi.org/10.7554/eLife.79530.sa1
Author response https://doi.org/10.7554/eLife.79530.sa2

# Additional files

## Supplementary files

• Supplementary file 1. Data collection and refinement statistics. *Information for the highest resolution shell is shown in parenthesis. †5.0% of the observed intensities were excluded from refinement for cross-validation purposes.

• Supplementary file 2. Small-angle X-ray scattering (SAXS) data collection and scattering-derived parameters for PKG Iβ 71–686.

• Supplementary file 3. Specific interactions between the R- and C-domains. Specific interactions between the R- and C-domains. The location of each residue within the complex is listed alongside of each amino acid. Ion pair, hydrogen-bond, and van der Waals interactions are notated as ↔, →, and ●—●, respectively.

• Supplementary file 4. Summary of the molecular dynamics (MD) simulations performed for PKG Iβ.

• MDAR checklist

## Data availability

Diffraction data have been deposited in PDB under the accession codes 7LV3 and 7MBJ.

The following datasets were generated:

| Author(s) | Year | Dataset title | Dataset URL | Database and Identifier |
|---|---|---|---|---|
| Sharma R, Lying Q, Casteel DE, Kim C | 2021 | Crystal structure of human protein kinase G (PKG) R-C complex in inhibited state | https://www.rcsb.org/structure/7LV3 | RCSB Protein Data Bank, 7LV3 |
| Kim JJ, Casteel DE, Kim C | 2021 | Crystal structure of cGMP dependent protein kinase I alpha (PKG I alpha) CNB-A domain with R177Q mutation | https://www.rcsb.org/structure/7MBJ | RCSB Protein Data Bank, 7MBJ |

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
