## [Editor Report]

This crystal structure of nearly full-length human cGMP-dependent protein kinase Iβ (PKG Iβ) provides convincing new insights into how in the absence of cGMP the activity of the catalytic domain is held in check by intramolecular interactions between both the upstream regulatory cGMP-binding domains and autoinhibitory segment and the catalytic domain, and how cGMP binding to the two cGMP-binding domains can relieve these inhibitory constraints leading to an increase in catalytic activity. The regulatory interactions in PKG Iβreveal similarities and differences to the way in which the cAMP-dependent protein kinase (PKA) regulatory domain inhibits the PKA catalytic subunit. The new structure of the activating PKG Iα R177Q CNB-A domain mutant, which resembles a cGMP-bound wild-type CNB-A domain, provides a nice explanation for how this point mutation activates PKG Iα and leads to the development of the TAAD (Thoracic Aortic Aneurysms and Dissections) syndrome.

---

## [Decision Letter]

**Decision letter after peer review:**

Thank you for submitting your article "An auto-inhibited state of protein kinase G and implications for selective activation" for consideration by *eLife*. Your article has been reviewed by 3 peer reviewers, including Tony Hunter as Reviewing Editor and Reviewer #1, and the evaluation has been overseen by Jonathan Cooper as the Senior Editor. The following individual involved in review of your submission has agreed to reveal their identity: Ping Zhang (Reviewer #2).

The three reviewers are in general agreement that your nearly-full-length human PKG1β structure is a significant advance in our understanding of the regulation of PKG by cGMP. No new experimental analysis is essential, although additional structure-function validation would strengthen your conclusions. However, all the reviewers agree that the paper is too long and difficult to read, and needs to be shortened and rewritten so that the main conclusions about how activity is held in check in the absence of cGMP, how cGMP binding activates PKG, and the extent to which the regulation of PKG is similar to and differs from that of the PKA C subunit by the R subunit and cAMP are more clearly articulated in each section. In addition, it would be useful to include some further discussion of why PKG is a dimer and whether there is functional crosstalk between the two monomers in a dimer, if negative regulation of kinase activity can be achieved through purely intramolecular interactions. In addition, a comparison of the human PGK1β structure with that of the full length *P. falciparum* PKG structure (doi: 10.1073/pnas.1905558116) would be informative. Some additional specific suggestions for improvement can be found in the individual reviews. We look forward to receiving a revised version.

Essential revisions:

The main essential revision is to undertake a major rewrite of the paper to shorten it and make clear the main conclusions in each section regarding the mechanisms of negative regulation of PKG kinase activity in the absence of cGMP and how it is activated upon binding cGMP, and how these mechanisms are similar to and differ from the regulation of the PKA C subunit by the R subunit and cAMP.

*Reviewer #1 (Recommendations for the authors):*

This paper is very long and was a hard slog to read through, and there may be a very limited number of people who will take the trouble to do this. The detailed discussion often obscures the main take home conclusion in each section. When the paper is rewritten and shortened to make the main conclusions clearer, perhaps some of the key interactions could be summarized in a graphic table instead of being described in detail in the text.

The functional consequences on PKG Iβ kinase activity and cGMP regulation of some additional mutations in key residues proposed to be involved in autoinhibition would strengthen the authors conclusions.

*Reviewer #2 (Recommendations for the authors):*

The manuscript will be very well-suited for publication in *eLife*, once the following comments are addressed.

– The cooperative nature of cGMP binding is mentioned in the introduction (line 58), however in the model (Figure 8), it is not clear of cGMP binding is cooperative only within a R/C dimer, or with two dimers in a full length R2/C2 tetramer. One approach would be to determine Hill coefficients to establish if the binding and activation by cGMP is cooperative in the tetramer.

It would be interesting to explain and develop further why PKG exists as a R-C:R-C homodimer and whether there is crosstalk between the two R-C monomers. One issue that is not addressed by the manuscript explicitly is the cooperativity of cGMP binding in full-length PKGIβ homodimer. The cooperative nature of cGMP binding is mentioned in the introduction (line 58), however in the model (Figure 8), it is not clear of cGMP binding is cooperative only within a R-C monomer, or with two R-C monomers in a full length R-C:R-C dimer. One approach would be to determine Hill coefficients to establish if the binding and activation by cGMP is cooperative in the R-C:R-C dimer

One general comment is that the manuscript is very long (over 50 pages when printed, 8 figures). Please consider streamlining the manuscript – you may consider condensing some of the discussion of the interfaces and moving parts of it to a supplementary material section or table.

*Reviewer #3 (Recommendations for the authors):*

I have no technical/experimental concerns with the work, it appears to be technically sound on all fronts and is an important advance suitable for publication in *eLife*. Suggestions for improving presentation:

1. I feel that contrast/compare with PKA is given short shrift. I want to know what is the same and what is different, and it is not easy to get that at a glance. That it looks almost the same as PKA (if indeed it does) does not make the present structure less interesting. Perhaps we could learn this right from the start, in the "overall structure" section?

2. The authors frequently "bury the lede" as they say in the newspaper trade. For example, the overall structure section starts out talking about individual domains and how they differ from previous structures of isolated domains. Actually, I am not sure they ever get around to describing the overall structure in this section, but there is a picture of it in Figure 1. Why not start with something like: In the autoinhibited state, the cyclic-nucleotide binding domains bind on either side of the kinase C-lobe, positioning the autoinhibitory segment to bind as a pseudosubstrate at the kinase active site. This overall organization is (essentially the same as/ similar to/ reminiscent of) that of protein kinase A….

3. In figure 1b, it is hard to get a sense of the structure of the kinase domain, and where on the kinase domain the regulatory domain is binding. At a minimum, labeling N- and C-lobes would help. Or perhaps a ribbon representation of the kinase would be better? Maybe with semi-transparent surface?

4. Would also be nice to get the comparison with PKA into Figure 1. (there is a nice superposition in Figure 3). Certainly up to the authors to decide what to describe/include in their manuscript, but I think it would be interesting and helpful to have a comparison of R:C contacts of PKA vs PKG, perhaps in the form of a sequence alignment with symbols to represent the various interdomain interfaces. What is conserved and what is unique to PKG?

5. Is there experimental evidence (beyond the prior crystal structure 4Z07) for dimerization of the CNB domain module in the nucleotide-bound state? If not, the text/figures describing the reorientation of these domains between autoinhibited and active (dimerized) states feels over-done.

---

## [Author Response]

Essential revisions:The main essential revision is to undertake a major rewrite of the paper to shorten it and make clear the main conclusions in each section regarding the mechanisms of negative regulation of PKG kinase activity in the absence of cGMP and how it is activated upon binding cGMP, and how these mechanisms are similar to and differ from the regulation of the PKA C subunit by the R subunit and cAMP.

We have significantly shortened the main text and attempted to state the regulation and activation mechanisms mediated by dynamic R-C interactions. Our main text includes now 6774 words (~35000 characters) reduced by ~1000 words (~4,000 characters) from the original submission.

We addressed why PKG is a dimer and its role in the cooperativity in the Discussion. We also provided our thoughts on functional crosstalk between the two monomers in a dimer in the same section. We also compared our PKG Iβ structure with the PKA R:C heterodimer structure and the full-length *P. falciparum* PKG structure at the overall structure section and provided RMSD values.

Reviewer #1 (Recommendations for the authors):This paper is very long and was a hard slog to read through, and there may be a very limited number of people who will take the trouble to do this. The detailed discussion often obscures the main take home conclusion in each section. When the paper is rewritten and shortened to make the main conclusions clearer, perhaps some of the key interactions could be summarized in a graphic table instead of being described in detail in the text.

We thank the reviewer for encouraging us to focus on the most important findings. We have relegated many of the details about R:C contacts to Supplementary file 3, which can be compared directly with Figure 3 of Kim et al., 2005, Science to appreciate similarities and differences between PKG and PKA auto-inhibition.

The functional consequences on PKG Iβ kinase activity and cGMP regulation of some additional mutations in key residues proposed to be involved in autoinhibition would strengthen the authors conclusions.

Our structure enabled us to design the KR/EE mutation and to provide structural interpretations for many mutational analyses performed by others over the last two decades, which we cite and discuss.

Reviewer #2 (Recommendations for the authors):The manuscript will be very well-suited for publication in eLife, once the following comments are addressed.– The cooperative nature of cGMP binding is mentioned in the introduction (line 58), however in the model (Figure 8), it is not clear of cGMP binding is cooperative only within a R/C dimer, or with two dimers in a full length R2/C2 tetramer. One approach would be to determine Hill coefficients to establish if the binding and activation by cGMP is cooperative in the tetramer.It would be interesting to explain and develop further why PKG exists as a R-C:R-C homodimer and whether there is crosstalk between the two R-C monomers.

We addressed the cooperativity of cGMP binding/activation of PKG Iβ by citing Smith et al., and stating that “auto-inhibited monomeric PKG Iβ leucine zipper deletion mutants are activated by cGMP without exhibiting cooperativity, so dimerization is dispensable for auto-inhibition but essential for cooperative activation (line 477, pg 18).” We further propose our thoughts on why PKG exists as a R-C:R-C homodimer by stating “If each monomer of a leucine zipper-mediated dimer has two bound cGMP, the PBC-A and bent interdomain helices of one monomer can associate with the same regions of the other monomer to which it is tethered and assemble into an antiparallel R:R interface (Figure 4A and 6D) (Kim et al., 2016). The formation of this interface further stabilizes the activated state (Figure 8, bottom right). This assembly helps sustain kinase activation since it occupies much of the R-domain surface that would otherwise be free to associate with the C-domain; it also shields the bound cGMP from solvent, which reduces the rate at which it might be released and degraded by phospho­diesterases (line 481-488, pp 18-19).”

One issue that is not addressed by the manuscript explicitly is the cooperativity of cGMP binding in full-length PKGIβ homodimer. The cooperative nature of cGMP binding is mentioned in the introduction (line 58), however in the model (Figure 8), it is not clear of cGMP binding is cooperative only within a R-C monomer, or with two R-C monomers in a full length R-C:R-C dimer. One approach would be to determine Hill coefficients to establish if the binding and activation by cGMP is cooperative in the R-C:R-C dimer

The cooperativity of cGMP activation for the full-length dimer is reported in Smith et al., and is cited in the revised manuscript, as is the lack of cooperativity for constructs lacking the LZ. We use these reported cooperativity data and our data showing the lack of inter-monomer effects on auto-inhibition to argue that cGMP cooperativity derives from R:R interactions in the activated state (see previous italicized paragraph).

This point was addressed in the responses to comments from reviewer 1.

Reviewer #3 (Recommendations for the authors):I have no technical/experimental concerns with the work, it appears to be technically sound on all fronts and is an important advance suitable for publication in eLife. Suggestions for improving presentation:1. I feel that contrast/compare with PKA is given short shrift. I want to know what is the same and what is different, and it is not easy to get that at a glance. That it looks almost the same as PKA (if indeed it does) does not make the present structure less interesting. Perhaps we could learn this right from the start, in the "overall structure" section?

We added sentences describing its similarity to PKA at the overall structure section as suggested (line 143 and line 147, pg 6).

2. The authors frequently "bury the lede" as they say in the newspaper trade. For example, the overall structure section starts out talking about individual domains and how they differ from previous structures of isolated domains. Actually, I am not sure they ever get around to describing the overall structure in this section, but there is a picture of it in Figure 1. Why not start with something like: In the autoinhibited state, the cyclic-nucleotide binding domains bind on either side of the kinase C-lobe, positioning the autoinhibitory segment to bind as a pseudosubstrate at the kinase active site. This overall organization is (essentially the same as/ similar to/ reminiscent of) that of protein kinase A….

This was addressed in the previous response.

3. In figure 1b, it is hard to get a sense of the structure of the kinase domain, and where on the kinase domain the regulatory domain is binding. At a minimum, labeling N- and C-lobes would help. Or perhaps a ribbon representation of the kinase would be better? Maybe with semi-transparent surface?

We modified Figure 1b so that a ribbon representation of the kinase is visible with semi-transparent surface. We also added labels for small (N) and large (C) lobes of the kinase domain, either termini, and the ordered residues at the R-C linker.

4. Would also be nice to get the comparison with PKA into Figure 1. (there is a nice superposition in Figure 3). Certainly up to the authors to decide what to describe/include in their manuscript, but I think it would be interesting and helpful to have a comparison of R:C contacts of PKA vs PKG, perhaps in the form of a sequence alignment with symbols to represent the various interdomain interfaces. What is conserved and what is unique to PKG?

We added sentences describing how R:C contacts in PKA are compared to PKG in overall structure section (line 147-149, pg 6), the R:C interface section, and the C-domain architecture and comparison with PKA section. We also emphasized how these differences might play important roles in minimizing cross-talk between the cyclic nucleotide signaling pathways in line 234, pg 9. We generated a R-C interaction table (Supplementary file 3), which can be compared directly with Figure 3 of Kim et al., 2005, Science to appreciate similarities and differences between PKG and PKA contacts that mediate auto-inhibition.

5. Is there experimental evidence (beyond the prior crystal structure 4Z07) for dimerization of the CNB domain module in the nucleotide-bound state? If not, the text/figures describing the reorientation of these domains between autoinhibited and active (dimerized) states feels over-done.

Discussing this structure is important because it provides a possible physical basis for cooperativity of cGMP binding; literature shows that the LZ is needed for cooperativity, and our data disprove the possibility that intermonomer interactions contribute to the auto-inhibited state.